# Marine-Derived Compounds with Anti-Alzheimer’s Disease Activities

**DOI:** 10.3390/md19080410

**Published:** 2021-07-24

**Authors:** Salar Hafez Ghoran, Anake Kijjoa

**Affiliations:** 1Department of Chemistry, Faculty of Science, Golestan University, Gorgan 439361-79142, Iran; s_hafezghoran@yahoo.com; 2Medicinal Plants Research Center, Yasuj University of Medical Sciences, Yasuj 75919-94779, Iran; 3ICBAS-Instituto de Ciências Biomédicas Abel Salazar and CIIMAR, Rua de Jorge Viterbo Ferreira 228, 4050-313 Porto, Portugal

**Keywords:** anti-Alzheimer’s disease, marine-derived compounds, acetylcholinesterase inhibitor, BACE-1 inhibition, anti-Aβ aggregation

## Abstract

Alzheimer’s disease (AD) is an irreversible and progressive brain disorder that slowly destroys memory and thinking skills, and, eventually, the ability to perform simple tasks. As the aging population continues to increase exponentially, AD has become a big concern for society. Therefore, neuroprotective compounds are in the spotlight, as a means to tackle this problem. On the other hand, since it is believed—in many cultures—that marine organisms in an individual diet cannot only improve brain functioning, but also slow down its dysfunction, many researchers have focused on identifying neuroprotective compounds from marine resources. The fact that the marine environment is a rich source of structurally unique and biologically and pharmacologically active compounds, with unprecedented mechanisms of action, marine macroorganisms, such as tunicates, corals, sponges, algae, as well as microorganisms, such as marine-derived bacteria, actinomycetes, and fungi, have been the target sources of these compounds. Therefore, this literature review summarizes and categorizes various classes of marine-derived compounds that are able to inhibit key enzymes involved in AD, including acetylcholinesterase (AChE), butyrylcholinesterase (BuChE), β-secretase (BACE-1), and different kinases, together with the related pathways involved in the pathogenesis of AD. The compounds discussed herein are emerging as promising anti-AD activities for further in-depth in vitro and in vivo investigations, to gain more insight of their mechanisms of action and for the development of potential anti-AD drug leads.

## 1. Introduction

Alzheimer’s disease (AD) is an irreversible and progressive neurodegenerative disorder affecting more than 50 million people across the world [1]. Although the pathogenesis of the disorder is complicated, it is directly related to the central nervous system (CNS) and generally emerges by a slow and irreversible progression of dementia, which is terminated with not only a decrease in cognition and memory, but also an alteration of languages, thinking, calculation, and personal behaviors [2]. The understanding of AD causality is still incomplete and a serious debate regarding whether AD is a singularly distinct form of dementia is ongoing [3]. Besides genetic and environmental factors [4,5], to date, several factors, including cholinesterase (acetyl- and butyrylcholinesterase) enzymes [6], proteases with β-site amyloid precursor protein cleaving cascade (β- and γ-secretases) [7], β-amyloid (Aβ) aggregation [3,8], protein kinase C [9], tau protein [3,10], glycogen-synthase kinase 3β (GSK-3β) [11], neurodegenerators [12], and oxidants [13] are recognized in this brain dysfunction. Moreover, metals, especially zinc, also have a key role in neuronal signaling and neurotransmission. However, whether this metal dyshomeostasis or aggregation in the brain is associated with AD is yet to be unraveled [14]. Given that AD is an age-related neurodegenerative disease and becomes even worse with time, searching for compounds from natural sources that can combat the causes or alleviate the symptoms of this disease is imperative. For example, epidemiological studies underlined that daily consumption of seaweed and ω-3 polyunsaturated fatty acids (PUFA) can slow down cognitive deficit in apolipoprotein E (apoE) carriers [15], which is in line with a meta-analysis conclusion that 0.1 g/day increment of dietary docosahexaenoic acid (DHA; **1**) intake was associated with lower risks of dementia [16]. Although some therapeutic agents have been described for alleviation of the AD symptoms in moderate-to-severe AD patients, a majority has been restricted to those that have mild efficiency and significant side effects [3]. These include the approved cholinesterase inhibitors [17], the noncompetitive *N*-methyl-D-aspartate (NMDA) receptor antagonists, memantine [18], and a combination of memantine with donepezil, as a cholinesterase inhibitor [19]. A number of marine-derived secondary metabolites have been revealed to possess neurological activities with different mode of actions. Fascaplysin (**2**), yessotoxin (**3**), bryostatin-1 (**4**), and saponins are among marine compounds whose mechanisms of action responsible for their neurological effects have been investigated [20]. Indeed, Kabir et al. and Martins et al. have reviewed various marine natural products, i.e., DHA (**1**), bryostatin-1 (**4**), anabaseine (**5**), homotaurine (**6**), and its derivatives, rifampicin (**7**), anhydroexfoliamycin (**8**), undecylprodigioisin (**9**), GV-971 (**10**), 13-desmethyl spirolide C (**11**), and dictyostatin (**12**), which have successfully passed the preclinical or clinical trials phases. These marine compounds show a well-tolerated behavior in AD individuals with no significant drug-associated side effects (Figure 1) [21,22]. Hence, many researchers have turned to the nature to search for bioactive compounds from terrestrial and marine resources, including plants, invertebrates, and microorganisms [23]. The fact that the marine environments have proved to be an important source of structurally unique compounds with an unprecedented pharmacological activity, much attention has been paid to compounds obtained from marine organisms [24]. Therefore, the aim of this work is to present a comprehensive review of marine-derived compounds that are able to either inhibit or block the key enzymes and the related pathways leading to AD as promising neuroprotective agents. For practicality, the compounds presented in this review are categorized according to their structural classes while their activities and mechanisms of action are discussed for each group of compounds or each individual compound. We have also ordered the marine resources starting from microorganisms, i.e., bacteria and fungi, to macroorganisms, i.e., tunicates, corals, sponges, and algae. The search engines used to search for the compounds in this review are PubMed, MEDLINE, Web of Science, and Scopus.

## 2. Marine-Derived Compounds That Inhibit Acetylcholinesterase (AChE) and Butyrylcholinesterase (BuChE) Activities

Although the main causes of AD are still not completely understood, a majority of researchers have focused on investigating the key enzymes that are involved in the progression of AD and other neurodegenerative diseases. Among them, cholinesterase enzymes have been considered relevant targets [25]. In order to facilitate a readability, the compounds discussed in this section are grouped according to their chemical classes.

### 2.1. Alkaloids

Marinoquinoline A (**13**), a pyrroloquinoline alkaloid, obtained from a new species of a marine gliding bacterium *Rapidithrix thailandica*, which was isolated from a decayed wood, collected from the Andaman Sea in Thailand, was shown to exhibit a potent AChE inhibitory activity, with an IC_50_ value of 4.9 μM (IC_50_ of galantamine = 0.6 μM). Interestingly, the two pyrrole derivatives 3-(2′-aminophenyl)-pyrrole (**14**) and 2,2-dimethylpyrrolo-1,2-dihydroquinoline (**15**) (Figure 2), obtained from *R. thailandica* strains TISTR 1742 and SH5.13.2, which were isolated from a submerged sea grass blade, collected in the Andaman Sea, and from a debris collected from the Gulf of Thailand, respectively, were inactive in the AChE-inhibition assay (enzyme inhibition < 30% at 0.1 g/L). The authors speculated that **15** was an isolation artifact of **14** [26].

The ethyl acetate (EtOAc) extract of the culture of *Acrostalagmus luteoalbus* TK-43, an endophytic fungus obtained from a marine green alga *Codium fragile*, collected in Turkey, furnished three pairs of enantiomers of *N*-methoxyindolediketopiperazine derivatives, acrozines A–C (**16**–**18**). The enantiomers of each pair were separated by a chiral high performance liquid chromatography (HPLC) to afford (+)-**16a**/(‒)-**16b**, (+)-**17a**/(‒)-**17b**, and (+)-**18a**/(‒)-**18b** (Figure 2). Compounds **16**–**18** were assayed for their anti-AChE activity by a slightly modified Ellman’s method. The results showed that the racemic mixture of **16**, i.e., (±)-**16a**/**16b** (IC_50_ = 9.5 µM) displayed anti-AChE activity ca. six-fold stronger than that of (±)-**17a**/**17b** (IC_50_ = 60.7 µM) and ca. 13-fold stronger than that of (±)-**18a**/**18b** (IC_50_ = 130.5 µM). Interestingly (+)-**16a** (IC_50_ = 2.3 µM) exhibited anti-AChE activity six-fold higher than that of its antipode while (+)-**17a** (IC_50_ = 78.8 µM) and (+)-**18a** (IC_50_ = 160.6 µM) exhibited weaker activities than those of their antipodes, respectively [27]. The alkaloid penicillamine (**19**) (Figure 2) was obtained as a racemic mixture from a marine-derived fungus *Penicillium commune* 366606, which was isolated from seawater, collected in China. Compound **19** was subjected to a HPLC separation using a chiral column to give (+)-**19a** and (‒)-**19b** (Figure 2), and the absolute configurations of their stereogenic carbons were established by comparison of experimental and calculated electronic circular dichroism (ECD) spectra. Compound (+)-**19a** exhibited more potent anti-AChE activity (32.4% inhibition) than its antipode (18.7% inhibition). However, both enantiomers were less active than the positive control, tacrine (43.6% inhibition) [28]. Pulmonarins A (**20**) and B (**21**) (Figure 2), isolated from the colonial ascidian *Synoicum pulmonaria,* which was collected off the Norwegian coast, exhibited reversible and noncompetitive inhibition of electric eel acetylcholinesterase (eeAChE) in vitro. Interestingly, although the structures of **20** and **21** are quite similar, **20** exhibited weaker inhibition (IC_50_ = 105 µM, and inhibitory constant, *Ki* = 90 μM) than **21** (IC_50_ = 36 µM, *Ki* = 20 µM). The *Ki* value of **21** (20 μM) indicates its moderate binding affinities for the vertebrate AChE, which is similar to that of the FDA-approved drug, galantamine (*Ki* = 2‒20 μM against AChE from different vertebrate species) [29].

Turk et al. isolated a pseudozoanthoxanthin analog **22** (Figure 2), from the ethanol extract of the zoanthid crust coral *Parazoanthus axinella* from underwater caves and crevices of the Mediterranean Sea. Compound **22** exhibited inhibition of eeAChE activity. The Dixon plot showed that **22** acted as a competitive inhibitor with a *Ki* value of 4 µM, and is therefore a moderate inhibitor [30]. Inspired by the anti-AChE activity of **22**, and from the observation that the 2-aminoimidazole group in this compound is also found in the known β-site amyloid precursor protein-cleaving enzyme 1 (BACE-1) inhibitors, Vitale et al. have searched for molecules that share a similar scaffold as potential multi-targeted compounds for AD therapy. Two compounds, i.e., 2-amino-3,9-dimethyl-5-methylamino-3*H*-1,3,4,6-tetrazacyclopent[e]azulene (**23**), a pseudozoanthoxanthin from an unidentified Caribbean zoanthid, and the bromopyrrole alkaloid stevensine (**24**) (Figure 2), recovered from the sponge *Axinella verrucosa,* which was collected in the Gulf of Naples, were selected for this study. A combined approach of molecular docking and molecular dynamics (MD) simulations suggested **23** and **24** as possible inhibitors of AChE, BuChE, and BACE-1, and that **23** can bind to both catalytic anionic site (CAS) and peripheral anionic site (PAS) of human acetylcholinesterase (hAChE), whereas **24** can only bind to CAS of hAChE. Moreover, modeling shows also a single binding mode for **23** to CAS of hBuChE while **24** loosely binds to hBuChE. In vitro inhibition assays for hAChE and equine BuChE (eqBuCHE) showed that **23** exhibited a lower inhibitory activity on hAChE than **24**, but stronger activity on eqBuChE than **24**, supporting modeling predictions, which demonstrated differential activity of **23** and **24** on the two targets [31]. Fascaplysin (**2**) (Figure 2) is a marine-derived bis-indole alkaloid, first isolated as a quaternary salt in 1988 from a marine sponge *Fascaplysinopsis* sp., collected in Fiji [32], and later from marine sponges *F. reticulata*, *Thorectandra* sp., *Hyrtios erecta*, and a tunicate *Didemnum* sp. [33]. Since several indole alkaloids have been reported as strong AChE inhibitors and memory function enhancers, Bharate et al. have synthesized **2** by a new synthetic routes, starting from commercially available precursor, tryptamine, and the easily synthesizable glyoxal, to evaluate its AChE activity against eeAChE and eqBuChE by Ellman’s method. Compared to the positive control, donepezil (IC_50_ for eeAChE = 0.09 μM, and for eqBuChE = 5.52 μM), **2** showed weaker inhibition on both eeAChE (IC_50_ = 1.49 μM) and eqBuChE (IC_50_ = 90.47 μM). Kinetic study revealed its noncompetitive inhibition indicating that **2** binds with high affinity to a site different from the catalytic one. A docking study revealed that the main interaction between **2** and AChE is a π–π interaction between the terminal aromatic ring of **2** and the Trp-279 residue of PAS. Therefore, the authors concluded that a lower potency of **2**, when compared to donepezil, can be attributed to lower hydrophobic fitting and π–π stacking interactions, fewer hydrogen bonding interactions, and no interaction with a key catalytic site residue Trp-84 [33]. In order to establish a structure–activity relationship of **2** for P-glycoprotein (P-gp) induction activity, Manda et al. synthesized a series of analogs of **2**, among which, 4,5-difluorofascaplysin (**25**) was found to inhibit 62% of eeAChE at 1 mM. Compound **25** also interacts only with the Trp-286 residue of PAS (corresponding residue in human AChE) by hydrophobic π–π interactions in a similar manner to **2**. Besides hydrophobic interactions, **2** also enters deeper into the catalytic pocket due to additional bulkier halogen groups on the E-ring, and exhibits a polar hydrogen bond with Gly-221 and H_2_O 737 at AChE catalytic site, thus preventing the entry of a substrate to the AChE catalytic site [34].

Due to multiple pathological mechanisms of AD, single-target drugs might have a limited efficacy. On the other hand, multi-target drugs could have superior activities when compared with single-target agents. Based on this hypothesis, Pan et al. [35] have synthesized nine fascaplysin analogs viz. 13-oxo-12,13-dihydropyrido[1,2-a:3,4-b′]diindol-5-ium bromide (**26**), 9-methyl-13-oxo-12,13-dihydropyrido[1,2-a:3,4-b′]diindol-5-ium bromide (**27**), 8-methyl-13-oxo-12,13-dihydropyrido[1,2-a:3,4-b′]diindol-5-ium bromide (**28**), 9-bromo-13-oxo-12,13-dihydropyrido[1,2-a:3,4-b′]diindol-5-ium bromide (**29**), 9-carboxy-13-oxo-12,13-dihydropyrido[1,2-a:3,4-b′]diindol-5-ium bromide (**30**), 9-carbamoyl-13-oxo-12,13-dihydropyrido[1,2-a:3,4-b′]diindol-5-ium bromide (**31**), 13-oxo-9-(phenylcarbamoyl)-12,13-dihydropyrido[1,2-a:3,4-b′]-diindol-5-ium bromide (**32**), 13-oxo-12,13-dihydropyrido[1,2-a:3,4-b′]diindol-5-ium chloride (**33**) and 9-methyl-13-oxo-12,13-dihydropyrido[1,2-a:3,4-b′]diindol-5-ium chloride (**34**) (Figure 3) to evaluate for several activities related to AD. Compounds **26**–**30** potently inhibited AChE and BuChE activities with IC_50_ values in µM range; however, their inhibition activities against AChE are stronger than those against BuChE. Interestingly, **27** inhibited AChE activity more potently than its parent compound **26**, suggesting that a modification of **2** could improve its cholinesterase inhibition effects. Curiously, **32** inhibited BuChE activity with an IC_50_ value of 2.90 μM, but did not inhibit AChE activity even at a concentration as high as 1 mM. Using the Lineweaver–Burk plot, it was found that the *Ki* value of **27** was 2.60 μM. Molecular docking analysis suggested that **27** might bind not only to the PAS residues, such as Tyr-341, Trp-286, Phe-338, and Tyr-337, through π–π interaction, but also to Trp-86 in CAS, through C-H and π interaction, leading to an enhanced AChE inhibition ability. For **28**, its 8-methyl group formed a hydrophobic interaction with the methylene group in the side chain of Trp-86, leading to a reduction of its inhibition of AChE activity. On the other hand, **29** and **30** could also bind to PAS and CAS of AChE; however, the electron-withdrawing bromine in **29** and the negatively charged carboxyl group in **30** have different levels of electron repulsion with the aromatic ring of Trp-86 in CAS, thus, leading to a reduced AChE inhibition ability of **29** and **30** compared with that of **27**. In the same manner, the electron repulsion between a negatively charged carbamoyl group of **31** and the aromatic ring of Trp-86 in CAS might significantly decrease its AChE inhibition capacity [35].

The *n*-BuOH fraction of the extract of a marine sponge *Petrosia* n. sp., collected in the intertidal zone of Phuket Island in Thailand, furnished petrosamine (**35**) and 2-bromoamphimedine (**36**) (Figure 3). Structural analysis revealed that **35** could exist in both keto (**35a**) and enol (**35b**) forms; however, the latter was detected only when D_2_O and CD_3_OD were used for NMR solvents. The AChE inhibitory activity assay, using AChE from the electric eel *Torpedo californica* (TcAChE), revealed that **35** exhibited potent AChE inhibitory activity (IC_50_ = 0.091 µM), which is about six times the potency of the reference compound galantamine (IC_50_ = 0.590 µM), whereas **36** showed only very weak potency, with IC_50_ value > 300 µM. ADocking study using two different crystal structures viz. 1DX6- and 1EVE-proteins of TcAChE, which are TcAChEs complexed with galantamine and donepezil, respectively, revealed that the calculated binding free energies for both systems of **35a** is approximately the same as those of **35b**, which are less than those of galantamine, whereas the free energies of **36** were the highest. The docking study results are in good agreement with their experimental IC_50_ values. Moreover, it also revealed direct interactions between galantamine, **35a**, **35b** and the two catalytic residues, Ser-200 and His-440, whereas **36** lacks an interaction with these two residues. Moreover, the docking data indicated that the quaternary dimethyl ammonium group in ring D of **35** donates major and strong interactions to the key amino acid residues located in the binding site (Glu-199 and Ser-200) of the enzyme [36]. The secondary metabolites from the marine sponges *Latrunculia biformis* and *L. bocagei*, obtained from coastal shelf environments around the Antarctic Peninsula, were also investigated for their anti-AChE activity. While the methanol extract of *L. biformis* furnished (+)-discorhabdins G (**37**) and (‒)-3-dihydro-7,8-dehydrodiscorhabdin C (**38**), the methanol extract of *L. bocagei* gave (+)-discorhabdins B (**39**) and L (**40**) (Figure 3), all belonging to the discorhabdin subclass, which is structurally characterized by an additional sulfur bridge between C-5 and C-8. These alkaloids were assayed for their inhibitory activities against eeAChE, recombinant hAChE and eqBuChE. All compounds tested effectively inhibited the activities of cholinesterases in a dose-dependent manner, with the highest activity against eeAChE. Moreover, **37** and **39** were found to be the most potent AChE inhibitors of the series, with IC_50_ values of 1.3 µM and 5.7 µM towards eeAChE, and 116 µM and 49.5 µM towards hAChE, respectively (physostigmine was used as a positive control with IC_50_ = 3 µM towards eeAChE, and 14.5 µM towards hAChE). The *Ki* values of **35** and **37** were 1.6 µM and 1.9 µM towards eeAChE, and 56.2 µM and 22.8 µM towards hAChE, respectively. On the other hand, **37** (IC_50_ = 7 µM) and **38** (IC_50_ = 15.8 µM) showed the strongest inhibition against eqBuChE among the tested compounds (IC_50_ of the reference compound, physostigmine salicylate, was 28.5 µM). Moreover, the kinetic of the inhibition of these cholinesterases by the tested compound, together with their analysis using Dixon plots, revealed that **37** and **39** act as reversible and competitive inhibitors through their binding to the active site of the free AChE or BuChE. The docking results for TcAChE showed that **37** had the lowest value of energies and the highest number of hydrophobic interactions, although with no specific hydrogen bonds, which is in agreement with the highest inhibitory activity observed. Moreover, the docking results for each of the tested compounds with hAChE showed higher energy and fewer hydrophobic interactions when compared with the inhibition of TcAChE. For the complex of **37** with hAChE, the computational data showed lower stability, fewer hydrophobic interactions and lack of hydrogen bonds when compared to the complex of **37** with TcAChE, thus supporting the experimental results that **37** had higher inhibitory activity against TcAChE than hAChE [37].

4-Acetoxyplakinamine B (**41**) (Figure 3), a stigmastane-type steroidal alkaloid, was isolated from the methanol extract of a marine sponge *Corticium* sp., collected in Southern Thailand. Compound **41** moderately inhibited eeAChE activity (IC_50_ = 3.75 μM) compared to the positive control, galantamine (IC_50_ = 0.59 μM). Kinetic study revealed an unusual mixed-competitive mode of inhibition of **41** [38]. The docking study indicated a strong potential for **41** to penetrate in and bind with the active subsites of AChE binding pocket. Simultaneous and favorable bonding interactions of **41** with PAS and anionic subsites (AS) provided a major clue behind its significant AChE inhibitory activity and mixed-competitive kinetic profile [39].

Sepcic et al. isolated polymeric pyridinium compounds (poly-APS), which are composed of two polymers of 29 and 99 repeating *N*-butyl(3-butylpyridinium) units (**42**) (Figure 3), from a marine sponge *Reniera sarai*, collected in the Northern Adriatic Sea. Poly-APS exhibited strong anti-AChE activity with an unusual pattern of inhibition, suggesting several mechanisms were involved. In order to have more insight of their AChE inhibitory activity and mechanisms, its monomeric unit, *N*-butyl-3-butylpyridinium iodide (**42**), was synthesized and evaluated for its anti-AChE activity. Kinetic study suggested that both polymeric and monomeric compounds inhibited AChE activity by irreversible and mixed-reversible patterns [40].

### 2.2. Phenolics

Five *p*-terphenyls, namely 6′-*O*-desmethylterphenyllin (**43**), 3-hydroxy-6′-*O*-desmethylterphenyllin (**44**), 3′′-deoxy-6′-*O*-desmethylcandidusin B (**45**), 3,3′′-dihydroxy-6′-*O*-desmethylterphenyllin (**46**), and 6′-*O*-desmethylcandidusin B (**47**) (Figure 4) were isolated from the EtOAc extract of cultures of a mangrove endophytic fungus *Penicillium chermesinum* ZH4-E2, isolated from the stem of the mangrove plant *Kandelia candel,* which was collected from the South China Sea. However, only **45** and **47** exhibited inhibitory activity against AChE with IC_50_ values of 7.8 and 5.2 μM, respectively (the positive control, huperzine A, showed an IC_50_ value of 0.12 μM) [41].

Yang et al. isolated nine depsidone derivatives, along with a cyclic peptide, from the EtOAc extract of cultures of a seaweed-associated fungus *Aspergillus unguis* DLEP2008001, collected from the South China Sea, and from the plasma-induced mutant of this fungus. Among the isolated depsidones, only aspergillusidone A (**48**) (Figure 4), isolated from the plasma-induced mutant, showed AChE inhibition with IC_50_ value of 56.8 μM (donepezil, a positive control, IC_50_ = 0.3 μM). In order to get an insight of the different anti-AChE activities between **48** and a structurally related analog, aspergillusidone G (**49**, IC_50_ > 102.4 μM), molecular docking studies were performed. Docking results showed that the docked pose of **48** with AChE has a negative CDOCKER energy of 7.86 kcal/mol and a negative CDOCKER interaction energy of 46.07 kcal/mol. Compound **48** also formed two hydrogen bonds between its 4-OH and Trp-84 (4.7 Å) as well as between its 10-COOH and Tyr-121 (6.5 Å). Moreover, electrostatic (Tyr-121, Trp-84, etc.) and van der Waals (Tyr-334, Phe-331, Phe-330, Ser-122, etc.) interactions also exist between **48** and AChE. All these interactions are known to be formed between potent AChE inhibitors and the enzyme active site. On the contrary, the docked pose of **49** with AChE has a negative CDOCKER energy of 10.02 kcal/mol and a negative CDOCKER interaction energy of 44.43 kcal/mol. Compound **49** mainly formed one hydrogen bond with AChE, i.e., between 4-OH and Tyr-70 (bond length 5.1 Å). However, electrostatic (Tyr-121, Ser-122, Trp-84, etc.) and van der Waals (Trp-84, Tyr-334, Phe-331, Phe-330, etc.) interactions exist between **49** and AChE [42]. The extracts of a culture broth and mycelia of a marine sponge-associate fungus *Talaromyces* sp. strain LF458, isolated from tissues of a marine sponge *Axinella verrucosa*, which was collected at Punta di Fetovaia, Isle of Elba (Mediterranean Sea) Italy, furnished, among other compounds, the anti-AChE metabolites viz. oxaphenalenone dimer, talaromycesone A (**50**), a penicillide derivative, AS-186c (**51**), and an isopentenyl xanthenone, talaroxanthenone (**52**) (Figure 4). Compound **52** showed the highest inhibitory activity toward AChE with IC_50_ value of 1.61 μM, followed by **51** with IC_50_ value of 2.63 μM, and **50** with IC_50_ value of 7.49 μM. The positive control, huperzine A, showed IC_50_ = 11.6 μM [43]. The methanol extract of a fermentation broth of a deep sea-derived fungus *Ochroconis* sp. FS449, isolated from a sediment collected from the Indian Ocean, yielded highly oxygenated usnic acid derivatives, ochuscins A–G (**53**–**59**) (Figure 4). Compounds **53**–**59** were evaluated for their anti-AChE activity by Ellman’s method, however, **54**–**56** and **59** exhibited moderate anti-AChE activity, with IC_50_ values ranging from 50 to 75 μM (a positive control neostigmine displayed IC_50_ value of 0.99 μM) whereas **53**, **57,** and **58** were inactive (IC_50_ > 200 μM) [44].

A dimeric polybrominated benzofuran, iantheran A (**60**) (Figure 5), isolated from a marine sponge *Ianthella* sp., which was collected from the Great Barrier Reef, Australia, inhibited the AChE activity with an IC_50_ value of 0.42 μM [45].

The aqueous methanol extract and organic solvent fractions of a brown alga *Ecklonia maxima* (Osbeck), collected on the West coast of South Africa, were investigated for their AChE inhibitory activity by Ellman’s method. The EtOAc fraction exhibited the most promising anti-AChE activity with IC_50_ value of 62.61 μg/mL. Purification of the EtOAc fraction by Sephadex LH-20 column, using mixtures of CHCl_3_/methanol, resulted in the isolation of phloroglucinol (**61**), together with two of its derivatives dibenzo[1,4]dioxine-2,4,7,9-tetraol (**62**) and eckol (**63**) (Figure 5). Compounds **62** and **63** exhibited anti-AChE inhibitory activity with IC_50_ values of 84.48 and 76.70 μM, respectively (IC_50_ of a positive control, galantamine = 1.8 μM) [46]. Moreover, 6,6ʹ-bieckol (**64**) (Figure 5), isolated form a red alga *Grateloupia elliptica*, which was collected offshore of Jeju island, Korea, showed moderate inhibitory activity toward AChE and BuChE compared to donepezil; however, its anti-BuChE activity (IC_50_ = 27.4 μM) was more potent than its anti-AChE activity (IC_50_ = 44.5 μM) [47].

Paudel et al. have evaluated the anti-AD activity of the methanol extract and the isolated compounds, including 2,3,6-tribromo-4,5-dihydroxybenzyl alcohol (**65**), 2,3,6-tribromo-4,5-dihydroxybenzyl methyl ether (**66**), and bis-(2,3,6-tribromo-4,5-dihydroxybenzyl) ether (**67**) (Figure 5) of the red alga *Symphyocladia latiuscula* (Harvey) Yamada, which was collected in Korean water, by using a Multiple Enzyme Targets approach. Compounds **65**–**67** inhibited both AChE and BuChE activities, being **67** the most potent inhibitor (IC_50_ values of 2.66 µM for AChE and 4.03 µM for BuChE), followed by **65** (IC_50_ values of 7.31 for AChE and 8.95 µM for BuChE) and **66** (IC_50_ values of 9.61 µM for AChE and 14.41 µM for BuChE) (the positive control, berberine, showed IC_50_ = 1.17 µM for AChE and 26.15 µM for BuChE). Kinetic study showed that **65**–**67** were mixed-type inhibitors of AChE and competitive inhibitors of BuChE, with *Ki* values of 0.58, 0.71, and 0.64 μM for AChE inhibition, and 1.15, 0.51, and 0.37 μM for BuChE inhibition, respectively. A docking study, using tacrine and donepezil as models, elucidates the binding mode of **65**–**67** in the active gorge of the *Tetronarce californica* AChE enzyme (1acj). Docking data showed that **65**–**67** were involved in the interaction with His-440 of CAS, in particular, **65** and **67** showed the hydrogen bond interaction with His-440 whereas **66** interacted with His-440 through the Br-O bond. Moreover, **65**–**67** also bound to PAS through interactions involving Tyr-334, Asp-72, Trp-84, and Tyr-121. In addition, **67** displayed more interactions with respect to the number of bromine atoms when compared to **65** and **66**. Thus, **65**–**67** showed mixed-type inhibition mode, binding to both CAS and PAS of AChE, with binding energies −7.47, −7.37, and −10.9 kcal/mol, respectively. In relation to BuChE, docking study showed catalytic inhibition of BuChE with low binding energies (−6.17, −6.44, and −9.80 kcal/mol, respectively). Compound **65** was involved in hydrogen bond interaction with Pro-285 and halogen bond interactions with His-438 (π−π T-shaped), Trp-231, Leu-286, Phe-329, and Gly-116. On the contrary, **66** displayed four hydrogen bond interactions with His-438, Ser-198, and Pro-285 (two hydrogen bonds), together with nonpolar interactions with Trp-231, Ala-199, Val-288, Leu-286, and Phe-329. On the other hand, **67**, which had the lowest binding energy among the three compounds and the reference compound, displayed multiple bond interactions with His-438 (two hydrogen bonds and one nonpolar interaction) and Trp-82 (three hydrogen bonds and one nonpolar interaction). Moreover, **67** also showed nonpolar interactions with Ala-328, Trp-430, Phe-329, and Gly-116. Taking together, **65**–**67** displayed competitive mode of BuChE inhibition through interactions with residues of CAS [48].

### 2.3. Terpenoids

#### 2.3.1. Sesquiterpenes

Li et al. isolated a polychiral bisabolane sesquiterpene, bisabolanoic acid A (**68**) (Figure 6), from a culture of a mangrove-associated fungus *Colletotrichum* sp. SCSIO KcB3-2, obtained from a mangrove plant *K. candel*, collected in Guangdong Province, China. Compound **68** showed a moderate inhibition of AChE activity with IC_50_ value of 2.2 μM (IC_50_ value of a positive control, huperzine A, is 0.30 µM). A docking study, using several crystal structures of AChE, showed that **68** was able to bind to the active pocket of different crystal structures of AChE, with Tyr-121 as the center of the binding site of AChE. Using the crystal structure 5EI5 of AChE, it was shown that the hydroxyl groups at C-7 and C-11 formed hydrogen bond interactions with the active site residue Tyr-121 and Ser1-22 of AChE while the carboxylic acid group also interacted with Arg-289 and Phe-288 by hydrogen bond. Consequently, it was deduced that OH-7, OH-11, and COOH-15 of **68** could play a key role for the inhibitory activity [49].

Onchidal (**69**) (Figure 6), a sesquiterpene containing an α/β-unsaturated aldehyde and an acetate group, was isolated from the mucous secretion of the mollusk *Onchidella binneyi*. Compound **69** inhibited AChE in a progressive and apparently irreversible manner. The apparent affinity of **69** for the initial reversible binding to AChE (*Kd*) was approximately 300 µM [50]. Stoddard et al., in their docking study of **69**, have found that **69** interacts with the esteratic site of AChE, which is consistent with Abramson’s work [50]. In the three poses generated for **69**, the cyclohexane ring is positioned in the bottleneck between the surface of AChE and the active site residues. In two of the three poses, the acetate ester is in the oxyanion hole. The lowest energy pose for **69** was generated from the 1DX6 receptor and places the acetate ester in the oxyanion hole, the aldehyde in the acyl pocket and the cyclohexane ring of **69** in the bottleneck created by Phe-330 and Tyr-121. The major interactions were potential hydrogen bond interactions with Gly-118 and Gly-119 of the oxyanion hole, hydrophobic contacts with the bottleneck and a possible hydrogen bond with His-440 of the catalytic triad. The irreversible inhibition of AChE by **69** makes it unsuitable for a direct use as an AChE inhibitor for human diseases since permanent inhibition of AChE leads to potentially deadly cholinergic toxicity. However, **69** could have a potential application in insecticides or pesticides. The interaction of **69** with the oxyanion hole is a new mechanistic area for incorporation into rational drug design of novel inhibitors [51].

With the aid of chemogenomics techniques, five halogenated sesquiterpenes viz. (−)-elatol (**70**), (+)-elatol (**71**), (+)-obtusol (**72**), (−)-dendroidol (**73**), and (−)-cartilagineol (**74**) (Figure 6), isolated from different populations of a red alga *Laurencia dendroidea*, collected off the coast of Brazil, were evaluated for their AChE inhibitory activity. Compounds **71** and **72** did not exhibit significant activities whereas **70**, **73,** and **74** were able to inhibit the enzyme with percent inhibition of 78.5%, 72.9%, and 61.3% respectively. Kinetic study of **70** confirmed its noncompetitive type of inhibition with *Ki* value of 255.9 μM. Molecular docking of **70** showed its interactions with important specific residues of the enzyme gorge involved in the activity of drugs used for treatment of AD. In particular, interactions between bromine and chlorine atoms with the center of the benzene ring of Tryp-86 and Tryp-286 may be contributing to the observed activity [52].

#### 2.3.2. Diterpenes

Anti-AChE activity-guided fractionation of the crude extract of a coral *Pseudoplexaura porosa* led to the isolation of 14-acetoxycrassine (**75**) (Figure 6). Anti-AChE activity assay by Ellman’s method of **75** and asperdiol (**76**) (Figure 6), a cembrane diterpene previously isolated from the sea whip *Eunicea knighti*, collected from the Colombian Caribbean [53] showed that both compounds exhibited a dose-dependent inhibition of AChE with IC_50_ values of 1.40 μM and 0.358 μM, respectively, which are less active than the positive control, galantamine (IC_50_ = 0.118 μM) [54].

Syad et al., in a continuation of their work on a red alga *Gelidiella acerosa* [55], have used a bioassay-guided fractionation of the benzene extract by column chromatography. The fractions showing anti-AChE and anti-BuChE activities were pooled and analyzed for its chemical constituents by LC-MS. Among several compounds detected, phytol (**77**) (Figure 6) was found to be a major component. Since **77** has been reported to have bioprotective potential, a pure phytol (**77**) was assayed for its anti-AChE and anti-BuChE activities. Compound **77** inhibited AChE and BuChE activities with IC_50_ values of 95.27 and 2.07 μg/mL, respectively. The molecular docking study showed that **77**, like the control ligands (rivastigmine, donepezil, galantamine) binds to Arg-277 at the active site of AChE [56].

#### 2.3.3. Sterols

The EtOAc extract of the culture of an endophytic fungus *Aspergillus flavus*, isolated from the inner tissue of a marine red alga *Corallina officinalis*, which was collected off the coast of Yantai, China, furnished 3β,4α-dihydroxy-26-methoxyergosta-7,24(28)-dien-6-one (**78**), episterol (**79**), (22*E*,24*R*)-ergosta-7,22-dien-3β,5α,6α-triol (**80**), (22*E*,24*R*)-ergosta-5,22-dien-3β-ol (**81**), and (22*E*,24*R*)-ergosta-4,6,8(14),22-tetraen-3-one (**82**) (Figure 6). Compound **78** was tested for AChE inhibitory activity, however, it showed only 5.5% inhibition at concentration of 100 μg/mL [57].

The petroleum ether fraction of the crude aqueous ethanol extract of a marine sponge *Xestospongia testudinaria*, collected from the coast of Sanya (South China Sea), Hainan province of China, furnished, among other compounds, cholesterol, 24-hydroperoxy-24-vinylcholesterol (**83**), saringosterol (**84**), 24-methylcholest-5-ene-3β,25-diol (**85**), and 29-hydroperoxystigmasta-5,24(28)-dien-3β-ol (**86**) (Figure 6). Compounds **83**–**86** were assayed for their AChE inhibitory activity by Ellman’s method, however, only **83** and **86** exhibited moderate anti-AChE activity with IC_50_ values of 11.45 and 14.51 μM, respectively (IC_50_ value of tacrine, a positive control, is 0.41 μM) whereas **84** and **85** did not show inhibition at concentrations up to 50 μM [58].

Castro-Silva et al. evaluated the in vitro and in silico AChE inhibitory activity of fucosterol (**87**) (Figure 6), isolated from the extract of a sea weed *Sargassum horridum*. Compared with the positive control, neostigmine, kinetic study showed that **87** is a noncompetitive hAChE inhibitor and has higher affinity than neostigmine for hAChE. Molecular docking results showed that both **87** and neostigmine are coupled to the AChE binding site, sharing a similar map of contacts. Most of the residues that stabilize the two complexes are hydrophobic residues with free energy values in picomolar and nanomolar range for **87** (−10.68 kcal/mol) and neostigmine (−7.02 kcal/ mol), respectively, suggesting that **87** not only has a higher affinity for AChE than neostigmine, but also acts as a noncompetitive hAChE inhibitor, which is different from a stabilization of galantamine (competitive hAChE inhibitor) [59].

#### 2.3.4. Tetraterpenes

Fucoxanthin (**88**) (Figure 7) is a natural carotenoid which is abundant in edible brown seaweeds and possesses a myriad of biological activities [60]. Lin et al. isolated **88** from *S. horneri* to evaluate its effects on cognitive impairment in vivo and its capacity to inhibit some key enzymes in vitro. Compound **88**, at 100 and 200 mg/kg, and donepezil significantly reduced the scopolamine-induced increase of AChE activity in the hippocampus of ICR mice. In an in vitro AChE activity assay, **88** was shown to directly inhibit AChE activity with an IC_50_ value of 81.2 µM. Moreover, kinetic study showed that **88** is a noncompetitive inhibitor of AChE, with *Ki* value of 42 µM for AChE. Molecular docking study reveals that **88** showed favorable interaction mainly with PAS of AChE by forming hydrogen bonds with Asp-283 and Ser-286 residues [61].

#### 2.3.5. Meroterpenoids

Asperpenes D (**89**) and E (**90**) (Figure 7) were obtained from the EtOAc extract of a marine-derived fungus *Aspergillus* sp. SCS-KFD66, isolated from a bivalve mollusk *Sanguinolaria chinensis*, which was collected from Haikou Bay, China. Compound **90**, at a concentration of 50 μg/mL, exhibited weak inhibitory activity against AChE, with an inhibition rate of 19.5% (the positive control, tacrine has an IC_50_ = 0.02 µM) [62]. Territrem B (**91**) (Figure 7), obtained from the extract of a liquid culture of a marine-derived fungus *A. terreus* BCC51799, isolated from a decayed wood sample from the Gulf of Thailand, exhibited anti-AChE activity with IC_50_ = 0.071 µM (IC_50_ of galantamine, a positive control = 0.097 µM) [63]. Nong et al. described the isolation of eight territrem derivatives and nine butyrolactone derivatives form the EtOAc extract of a solid culture of a marine-derived fungus *A. terreus* SCSGAF0162, isolated from a gorgonian coral *Echinogorgia aurantiaca*, which was collected in the South China Sea. The isolated compounds were assayed for their anti-AChE activity by Ellman’s method, however, only **91**, territrems C (**92**), D (**93**), and E (**94**), arisugacin (**95**), and terreulactone C (**96**) (Figure 7) exhibited anti-AChE activity with IC_50_ values of 4.2, 20.1, 4.2, 4.5, 11.9, and 50 nM, respectively (the positive control huperzine A showed IC_50_ = 39.3 nM) [64]. Ding et al. [65] evaluated anti-AChE activity of five α-pyrone meroterpenoids, **91**, **96**, 3-epiarigsugacin E (**97**), arisugacin B (**98**) and arisugacin D (**99**) (Figure 7), purified from the EtOAc extract of the culture of an endophytic fungus *Penicillium* sp. SK5GW1L, which was isolated form leaves of a mangrove tree *K. candel*. Compound **96** exhibited potent inhibitory activity with IC_50_ = 0.028 µM, while **94** (IC_50_ = 0.23 µM) and **98** (IC_50_ = 3.03 µM) exhibited moderate inhibitory activity (IC_50_ of the positive control, huperzine A = 0.036 µM). On the other hand, **97** (IC_50_ = 38.23 µM) and **99** (IC_50_ = 53.39 µM) showed weak activity [65].

Two plastoquinones, namely sargaquinoic acid (**100**) and sargachromenol (**101**) (Figure 7), were isolated from the methanol extract of a brown alga *Sargassum sagamianum,* collected from Korean waters. Compounds **100** and **101** exhibited moderate anti-AChE activity with IC_50_ values of 23.2 and 32.7 μg/mL, respectively (IC_50_ values of the positive controls, tacrine and donepezil are 0.05 and 0.002 μg/mL, respectively). Compounds **100** and **101** also showed anti-BuChE activity, with IC_50_ values of 0.026 and 7.3 μg/mL, respectively (IC_50_ value of tacrine is 0.002 μg/mL). Interestingly, **100** showed particularly potent BuChE inhibitory activity which is 1000-fold greater than that of AChE [66]. Seong et al. have also isolated **100**, **101**, and sargahydroquinic acid (**102**) (Figure 7) from the hexane fraction of *S. serratifolium*, collected along the coast of Busan, South Korea. Compounds **100**–**102** showed moderate AChE inhibitory activity with IC_50_ values of 69.3, 97.3, and 124.3 µM, respectively (IC_50_ of berberine, a positive control = 1.6 µM), but more potent inhibitory activity against BuChE, with IC_50_ values of 10.5, 9.4, and 15.2 µM, respectively (IC_50_ of berberine, a positive control = 9.4 µM) [67].

### 2.4. Marine Fatty Acids

The (8*E*,12*Z*)-10,11-dihydroxyoctadeca-8,12-dienoic acid (**103**) (Figure 8), isolated from the EtOAc extract of the endophytic fungus *A. flavus*. exhibited a weak AChE inhibitory activity, with inhibition rate of 10.3% at a concentration of 100 μg/mL) [57].

The petroleum ether fraction of the crude extract of a marine sponge *Xestospongia testudinaria* furnished unsaturated fatty acids and their esters viz. xestospongic acid (**104**), its methyl ester (**105**), and 18-brornooctadeca-(9*E*,17*E*)-diene-5,7,15-triynoic acid (**106**) (Figure 8). Compound **104** displayed weak anti-AChE activity (IC_50_ = 12.65 µM) while **105** and **106** did not inhibit AChE activity at concentration as high as 50 µM [58].

Yang et al. examined the anti-AChE potency of a functional oil extracted from an edible brown seaweed *Hizikia fusiforme* and its fatty acid components. The functional oil showed an anti-AChE activity with IC_50_ value of 1 mg/mL. GC-MS analysis of the functional oil revealed the presence of oleic acid (**107**), palmitic acid (**108**), myristic acid (**109**), arachidonic acid (**110**), 11,14,17-eicosatrienoic acid (**111**) (Figure 8), and phytol (**77**). Anti-AChE activity assay of commercial pure samples of these compounds revealed that the anti-AChE activity of the original oil mainly originated from **110** (IC_50_ = 0.78 mg/mL) and **111** (IC_50_ = 0.50 mg/mL). Kinetic study showed that both **110** and **111** displayed a noncompetitive inhibition mode with *Ki* values of 5.70 and 0.84 mM/mL, respectively. A docking study showed that **110** combined with the active pocket of AChE and the docked pose of **110** and AChE showed a low CDOCKER interaction energy (−26.33 kcal/mol) and the carbonyl group of **110** forms two hydrogen bonds with Arg-220 and Arg-221. On the other hand, **111** combined to a site with the α-helical motif quite far from the active pocket. However, its docked pose showed even lower CDOCKER interaction energy (−50.36 kcal/mol) when compared to **110**, thus explaining its stronger anti-AChE activity [68].

### 2.5. Peptides

A cyclopentapeptide, *cyclo*-(L-Phe-L-Leu-L-Val-L-Leu-L-Leu) (**112**) (Figure 8) was obtained from a crude ethanol extract of the culture of a marine-derived fungus *Biscogniauxia mediterranea* strain LF657, isolated from a sediment collected in Eastern Mediterranean Sea. Compound **112** exhibited anti-AChE activity with a minimal inhibition concentration (MIC) of 5.87 μM. Huperzine A, the positive control, displayed IC_50_ value of 0.012 μM [69].

### 2.6. Miscellaneous Compounds

Among four brominated aliphatic hydrocarbons, isolated from a crude extract of a marine sponge *X. testudinaria*, mutafuran H (**113**) (Figure 8) showed significant in vitro inhibition of AChE enzyme with IC_50_ = 0.64 mM [58].

A glycoprotein with a molecular weight of approximately 10 kDa, consisting of carbohydrate (42.53%) and protein components (57.47%), was isolated from an edible brown macroalgae *Undaria pinnatifida* Harvey, purchased from the market in Korea [70]. This glycoprotein inhibited AChE and BuChE activities with IC_50_ values of 63.56 and 99.03 µM, respectively [71].

## 3. Marine-Derived Compounds That Inhibit β-Secretase (BACE-1) and γ-Secretase Activities

The β-amyloid (Aβ) peptide is widely considered to play a critical role in the pathogenesis of AD. Aggregates of Aβ are generated by cleavage of the amyloid precursor protein (APP) by β-secretase (BACE-1), followed by γ-secretase. BACE-1 cleavage is limiting for the production of Aβ, making it a particularly good drug target for inhibitors that lower the amount of soluble Aβ, preventing and/or reversing their aggregation. Interestingly, some marine-derived compounds have been proved to inhibit BACE-1 and, thus, preventing Aβ aggregation [72].

### 3.1. Alkaloids

Lodopyridones B (**114**), C (**115**), and A (**116**) (Figure 9), all possessing a thiomethyl-substituted 4-pyridone, thiazole, and chloroquinoline moieties, were purified from a culture broth of a marine-derived actinomycete *Saccharomonospora* sp. CNQ-490, isolated from a sediment sample collected from the La Jolla Submarine Canyon in La Jolla California. Compounds **114**–**116** showed weak inhibition of BACE-1 at 51%, 42%, and 60% at a concentration of 100 μM, respectively (IC_50_ of isoliquiritigenin, a positive control = 51.5 μM). Moreover, **116** exhibited a mild, but dose-dependent, inhibition on the release of Aβ_40_ and Aβ_42_ as their levels were reduced up to 41% and 49% of the control by **116**, at 1 mM, respectively, while isoliquiritigenin, at 51.5 µM, decreased their level to 48% and 50% of the control [73].

3-[(2,4-Dimethoxy)benzylidene]-anabaseine (DMXBA; GTS-21; **117**), a nicotinic acid-derived alkaloid, isolated from the mucus secretions of a marine ribbon warms *Amphiporus* sp., showed an agonist activity on alpha7 nicotinic acetylcholine receptors (α7-nAChRs) [74]. Takata et al. have examined an in vitro and in vivo γ-secretase activity using human neuroblastoma SH-SY5Y cells and APdE9 mice, respectively. In vitro results showed that pretreatment of SH-SY5Y cells with **117** (1–100 μM) significantly suppressed γ-secretase activity in cell fractions in a dose-dependent manner via downregulating presenilin 1, when compared with (3,5-difluorophenylacetyl)-Ala-Phg-Obut (DAPT; a γ-secretase inhibitor) that directly and reversibly inhibited γ-secretase activity. On the other hand, **117** attenuated in vivo Aβ aggregation via suppression of γ-secretase activity (treatment of APdE9 mice with 1 or 5 mg/kg DMXBA) [75].

Ianthellidone F (**118**), a pyrrolidone alkaloid, and lamellarins O (**119**), O1 (**120**), and O2 (**121**) (Figure 9), were isolated from the extract of a marine sponge *Ianthella* sp. CMB-01245, which was collected in Southern Australia. Compounds **118**, **119**, and **121** moderately inhibited BACE-1 activity with an IC_50_ value > 10 μM (40% inhibition at a concentration of 10 μM) whereas **120** showed a stronger activity (IC_50_ < 10 μM; 60% inhibition at a concentration of 10 μM). Although their levels of BACE-1 inhibition are modest, the ianthellidones and lamellarins can be considered as potential scaffolds for a development of BACE-1 inhibitor [76]. The same research group has further explored the extract of the same sponge and has isolated five dictyodendrins, i.e., dictyodendrins F (**122**), G (**123**), H (**124**), I (**125**), and J (**126**) (Figure 9), a rare class of marine alkaloids belonging to the lamellarin and ianthellidone structural classes. Compounds **122**, **124**–**126** exhibited significant BACE-1 inhibitory activity, with IC_50_ values ranging from 1 to 2 μM whereas **123** was inactive. Interestingly, **122**–**125** exhibited cytotoxicity against human colon cancer cell line SW620, being **123**, the 10-OMe analog, the most cytotoxic. On the other hand, **126** that possesses a *seco*-carbon skeleton and an unusual 1,2-diketone functionality did not exhibit cytotoxicity, which is worth further investigation [77].

Thirteen bromotyrosine-derived alkaloids, including purpuramines G (**127**), M (**128**), N (**129**), araplysillins II (**130**), IV (**131**), VII (**132**), VIII (**133**), IX (**134**), X (**135**), and XI (**136**), hexadellin A (**137**), purpurealidin I (**138**), and aplysamine 4 (**139**) (Figure 10) were isolated form the methanol extract of an Indonesian marine sponge *Aplysinella strongylata*. Compounds **127**, **128**, **130**, **132**, **134**–**139** showed moderate BACE-1 inhibition with IC_50_ values of 48.3, 42.0, 31.8, 39.6, 41.9, 31.4, 30.6, 41.7, 42.0, and 42.6 μM, respectively (IC_50_ of β-secretase inhibitor IV = 0.021 μM) [78].

2-Amino-3,9-dimethyl-5-methylamino-3*H*-1,3,4,6-tetrazacyclopent[e]azulene (**140**), a pseudozoanthoxanthin from an unidentified Caribbean zoanthid, and the bromopyrrole alkaloid, stevensine (**141**) (Figure 10), recovered from a marine sponge *A. verrucosa,* which was collected in the Gulf of Naples, were found to inhibit the murine BACE-1 with IC_50_ values of 0.9 and 1.4 µM, respectively. Molecular docking and MD studies showed that both **140** and **141** possibly bind to both “open” and “closed” states of hBACE-1 active site [30].

### 3.2. Phenolics

Compound **64** (Figure 5), isolated form the alga *G. elliptica*, inhibited BACE-1 activity with 18.6% inhibition at a concentration of 1 µM (the standard reference, Z-Val-Leu-Leu-CHO, showed 75% inhibition at 1 µM) [47], while **66** and **67**, isolated from the red alga *S. latiuscula* (Harvey) Yamada, also exhibited significant inhibition of BACE-1 activity with IC_50_ values of 5.16, 4.79, and 2.32 µM, respectively. Interestingly, the BACE-1 inhibition potential of the three compounds was 5-10-fold higher than the reference drug, quercetin (IC_50_ = 25.21 μM). In order to better understand the inhibitory mechanism of BACE-1, the interactions of **65**–**67** with the BACE-1 structure were evaluated by molecular docking simulation. The results showed that **65** and **67** bound to the active allosteric site of BACE-1 with low energies (−6.59 and −5.98 kcal/mol, respectively). However, these interacting residues are not involved in the binding of **67** to the enzyme. The determining factors for competitive inhibition by **67** were the interactions with the catalytic Asp dyad (Asp-32−Asp-228), Ser-36, Ile-126, and Gly-230), which were not observed for **65** and **66** [48].

The chlorinated phenolic metabolites, saccharochlorines A (**142**) and B (**143**) (Figure 11), isolated from the saline culture of a marine-derived bacterium *Saccharomonospora* sp. KCTC-19160, exhibited weak inhibition of BACE-1 activity (42.4% and 32.0% inhibition, respectively) at a concentration of 50 μM (a positive control, isoliquiritigenin showed 56.7% inhibition at 50 μM). Moreover, treatment of both compounds did not cause any increase in levels of Aβ_40_ and Aβ_42_ in the SH-SY5Y neuroblastoma cells. On the contrary, both compounds caused an increase in Aβ_40_ and Aβ_42_ levels in a dose-dependent manner in H4-APP neuroglial cells, therefore exhibiting a stimulatory effect on the release of Aβ_40_ and Aβ_42_ in H4-APP cells. The authors suggested that the contradiction between the biochemical BACE-1 inhibitory assay and the cell-based BACE-1 inhibitory assay might be due to a difference in the efficiency of BACE-1 in cleaving APP, which depends on cellular models and subcellular environments. This was supported by the fact that isoliquiritigenin, a BACE-1 inhibitor, elevated Aβ secretion in H4-APP culture media and C-terminal fragment-β (CTF-β) concentration in the cells, respectively [79].

Williams’s group described the isolation of two pentacyclic diketones, named xestosaprols D (**144**) and E (**145**) (Figure 11), from the extract of a marine sponge *Xestospongia* sp., collected from the Indonesian waters. Compound **144** showed weak inhibition of BACE-1 activity with IC_50_ value of 30 μg/mL [80]. The same research group has further explored the chemistry of the marine sponge *Xestospongia* sp., collected at the coral reef from the Indonesian waters. The EtOAc-soluble fraction of its crude extract furnished eight xestosaprol analogs, named xestosaprols F–M (**146**–**153**) (Figure 11). Compounds **146**–**153** exhibited a moderate dose-dependent inhibition of BACE-1 with IC_50_ values ranging from 82 to 163 μM (IC_50_ of the positive control, the secretase inhibitor IV = 0.015 μM) [81].

Fucofuroeckol-b (**154**) (Figure 11), an eckol derivative, was obtained by bioassay-guided isolation from the EtOAc fraction of the crude ethanol extract of a brown alga *Eisenia bicyclis*, collected from the coast of South Korea. Compound **154** showed a strong β-secretase inhibition with IC_50_ value of 16.1 μM. Kinetic study revealed a noncompetitive inhibition pattern of **154** in the Dixon plot, with *Ki* value of 10.1. Compound **154** was also attenuated Aβ-induced cytotoxicity in SH-SY5Y cells, probably by inhibition of β-secretase-mediated downregulation of Aβ. Moreover, the western blot results indicated that treatment with **154** decreased soluble-APPβ (sAPPβ) and Aβ_42_ expression in a dose-dependent manner in SH-SY5Y cells overexpressing APP695swe (SH-SY5Y-APP695swe cells) [82].

### 3.3. Terpenoids

#### 3.3.1. Sesquiterpenoids

Several sesquiterpenoid quinones, isolated from the extracts of a marine sponge *Dactylospongia elegans*, were assayed for their inhibition of BACE-1 activity. However, only ilimaquinone (**155**) and smenospongorine (**156**) (Figure 12) showed moderate inhibition of BACE-1 activity with IC_50_ values of 65 and 78 μM, respectively (β-secretase inhibitor IV was used as a reference inhibitor) [83].

#### 3.3.2. Diterpenoids

In the course of investigation of the effects of gracilin diterpenoids, isolated from a marine sponge *Spongionella* sp., for their capacity to intervene with processes involved in the pathology of AD, Leiros et al. have evaluated the in vitro capacity to inhibit the enzyme β-secretase of gracilins A (**157**), H (**158**), L (**159**), and tetrahydroaplysulphurin-1 (**160**) (Figure 12) by using a Free Resonance Energy Transfer (FRET) assay kit. The results showed that only **159** at 1 mM was able to produce a significant inhibition of BACE-1, decreasing its activity by 24.6%. Although **157** and **160** also reduced BACE-1 activity at 1 mM, their inhibition was not in a significant manner. Treating retinoic acid differentiated neuroblastoma BE(2)-M17 cells with **157**–**160**, at concentrations of 1 and 0.1 mM for 24 h, did not have any effect on Aβ_42_ levels. In addition, the levels of Aβ_42_, hyperphosphorylated tau, and extracellular signal-regulated kinase (ERK) at Ser-202 and Thr-205 were reduced after treatments with gracilins in vitro [84].

#### 3.3.3. Meroterpenoids

Sargaquinoic acid (**100**), sargachromenol (**101**), and sargahydroquinic acid (**102**) (Figure 7) also showed inhibition of BACE-1 activity with IC_50_ values of 4.4, 7.0, and 12.1 µM, respectively (the positive controls, quercetin and STA-200 showed IC_50_ values of 5.6 and 0.20 µM, respectively). Kinetic study showed a mixed-type inhibition of BACE-1 by **100** and **102** with *Ki* values of 4.0 and 1.6 mM, respectively, whereas **101** exhibited a noncompetitive inhibition with *Ki* value of 2.9 mM. Molecular docking study revealed that the three compounds not only formed a number of strong hydrogen bonds with several important amino acid residues located in the catalytic and allosteric sites of BACE-1, but also possessed a number of hydrophobic interactions with BACE-1, thus explaining their potency against this enzyme [67].

Lopez-Ogalla et al. used prenylated hydroxybenzoic acid (**161**) (Figure 12), isolated from a marine sponges *Spongia officinalis*, *Ircinia spinulosa, I. muscarum*, collected at different locations in the Mediterranean Sea, as a scaffold to design BACE-1 inhibitor. They have found that both natural and synthetic **161** inhibited Aβ_1–40_ production in a micromolar range. The percent inhibition of natural **161** was found to be 10% at 1 µM and 47% at 10 µM. Further studies on its mechanism of action showed that natural **161** also inhibited BACE-1 activity with a percent inhibition of 45% at 1 µM. Interestingly, the synthetic version showed slightly lower inhibitory activities for both Aβ formation and BACE-1 [85].

#### 3.3.4. Steroids

FRET-based BACE-1 inhibitory activity-guided fractionation of the ethanol extract of *Urechis unicinctus* (Phylum Annelida), also known as innkeeper worm, which was purchased from the market in Korea, led to the isolation of, among other steroids, hecogenin (**162**), and cholest-4-en-3-one (**163**) (Figure 12). Both compounds inhibited BACE-1 activity with EC_50_ value of 116.3 and 390.6 μM, respectively. The positive control, curcumin, showed 88.03% inhibition at a concentration of 0.5 mg/mL [86].

#### 3.3.5. Carotenoids

Compounds **87** (isolated from a seaweed *E. stolonifera*) and **88** (isolated from the brown algae *Undaria pinnatifida* and *E. bicyclis*) (Figure 6) were found to exhibit an in vitro BACE-1 inhibitory activity with IC_50_ values of 64.12 and 5.31 µM, respectively (IC_50_ of quercetin, a positive control = 10.19 µM). Kinetic study showed that **88** exhibited a mixed-type inhibition with *Ki* = 7.19 µM, whereas **87** showed a noncompetitive inhibition with *Ki* = 64.59 µM. Molecular docking study predicted that ligand interactions of **88** in the active site of BACE-1 consisted of two hydrogen bonds interaction between Gly-11 and Ala-127 residues of the enzyme. Moreover, two hydroxyl groups of **88** and eight residues viz. Thr-231, Tyr-198, Thr-232, Tyr-71, Ile-110, Ile-118, Leu-30, and Ile-126 of the enzyme participated in hydrophobic interactions with the methyl group of **88**. On the other hand, ligand interactions of **87** in the active site of BACE-1 consisted of one hydrogen bond interaction between Lys-224 of BACE-1 and one hydroxyl group of **87**, while seven residues, i.e., Ile-118, Tyr-71, Ile-226, Thr-231, Val-332, Phe-108, and Val-69 of the enzyme participated in hydrophobic interactions with the methyl group of **87**. Interestingly, the ligand interactions of quercetin in the active site of BACE-1 consisted of six hydrogen bonds interaction between Val-69, Trp-76, Ser-35, Asn-37, Ser-36, and Ile-126 residues of the enzyme and four hydroxyl groups and one oxygenated carbon of quercetin, whereas the residues Tyr-71 and Phe-108 of the enzyme participated in hydrophobic interactions with the methyl group of quercetin. Additionally, the binding energies of **87** and **88** are −10.2 and −7.0 kcal/mol, respectively, indicating that additional hydrogen bond might stabilize the open form of the enzyme and potentiate tighter binding to the active site of BACE-1, resulting in more effective BACE-1 inhibition [87].

### 3.4. Peptides

Tasiamide B (**164**) (Figure 13), a linear depsipeptide containing eight amino acid residues, was isolated from a marine cyanobacterium *Symploca* sp. [88]. Liu et al. [89] noticed that the presence of a statin-like unit (4-amino-3-hydroxy-5-phenylpentanoic acid, AHPPA) in the structure of **164** might have an affinity to the active site of BACE-1. Compound **164** and modified peptides with a central statin-core unit, which are characteristic of aspartic protease inhibitors, were synthesized and evaluated for BACE-1 inhibitory activity. The synthetic **164** exhibited inhibition of BACE-1 with IC_50_ = 353 nM (IC_50_ of the positive control, β-secretase inhibitor IV = 250 nM). By measuring secreted amounts of Aβ peptides formed upon sequential action of β- and γ-secretases in CHO 2B7 cells (clonal expression of APP695 wt), it was found that the activity of **164** was in a micromolar range, and thus exhibiting potency approximately 10–100-fold lower than in the enzymatic BACE-1 assay [89]. Tasiamide F (**165**) (Figure 13), an analog of **164**, was isolated together with **164** from the EtOAc fraction of the crude extract of a marine cyanobacterium *Lyngbya* sp., collected from patch reefs in Cocos Lagoon, Guam. Compound **165** inhibited BACE-1 activity (IC_50_ = 690 nM), which is approximately 8-fold less potent than **164** (IC_50_ = 80 nM) [90].

Lee et al. isolated β-secretase inhibitory peptide, with IC_50_ = 24.26 μM, from an enzymatic hydrolysate by neutrase enzyme of a skate (*Raja kenojei*) skin, obtained from a local skate processing plant in Korea. The amino acid sequence of this peptide was identified by MS/MS as Gln-Gly-Thr-Arg-Pro-Leu-Arg-Gly-Pro-Glu-Phe-Leu (**166**) (Figure 13), with a molecular weight of 1391 Da [91]. In another study, Lee et al. have used various enzymes to hydrolyze the muscle of sea hare (*Aplysia kurodai*) and have found that the trypsin hydrolysate had the highest β-secretase inhibitory activity compared to other hydrolysates. Purification of the hydrolysate by Sephadex G-25 column chromatography and HPLC using a C18 column, followed by amino acid sequence analysis by MS/MS revealed the structure of the active peptide as Ala-Ala-Leu-Met-Leu-Phe-Asn (**167**) (Figure 13). The peptide **167** exhibited IC_50_ = 74.25 μM, and acts as a competitive inhibitor against β-secretase [92]. A peptide with amino acid sequence Ser-Leu-Ala-Phe-Val-Asp-Asp-Val-Leu-Asn (**168**) (Figure 13), purified from a protein hydrolysate of Pacific hake (*Merluccius productus*) by Sephadex G-25 column chromatography and ODS-C18 reversed-phase HPLC, exhibited potent β-secretase inhibitory activity (IC_50_ = 18.65 μM) in SH-SY5Y cells stably transfected with the human ‘‘Swedish’’ amyloid precursor protein (APP) mutation APP695 (SH-SY5YAPP695swe) [93].

Among the hydrolysates obtained from hydrolysis of blackfin flounder (*Glyptocephalus stelleri*) muscle protein with various commercial proteases, the Alcalase hydrolysate exhibited the highest β-secretase inhibitory activity compared with other hydrolysates. Purification of this hydrolysate by Sephadex G-25 column chromatography and reversed-phase HPLC using an ODS column, followed by amino acid sequence analysis by electrospray ionization MS/MS led to the characterization of an inhibitor as a pentapeptide Leu-Thr-Gln-Asp-Trp (**169**) (Figure 13) with a molecular mass of 526.7 Da. This peptide showed an IC_50_ = 126.93 μM [94].

### 3.5. Polycyclic Ethers

Alonso et al. examined the effect of Gambierol (**170**), a polycyclic ether toxin produced by a dinoflagellate *Gambierdiscus toxicus*, and its synthetic heptacyclic (**171**) and tetracyclic (**172**) analogues (Figure 13) in an in vitro model of AD, obtained from triple transgenic (3xTg-AD) mice that express Aβ accumulation and tau hyperphosphorylation. They have found that **170**–**172** decreased both intra- and extracellular Aβ levels and tau hyperphosphorylation via modulation of *N*-methyl-D-aspartate (NMDA) receptors that is possibly secondary to voltage-gated potassium (K_v_) channel inhibition in an in vitro mouse model of AD [95]. Furthermore, both **170** and **172** displayed a dose-dependent inhibition of BACE-1 activity, falling to 52.25% with **170** and to 58.04% with **172**, at 10 µM [96].

## 4. Marine-Derived Compounds That Inhibit Aβ Aggregation

Overexpression of BACE-1 is a key factor for an increase of Aβ peptides generation as well as BACE-1-cleaved APP fragments. In addition, it results in an imbalance between Aβ production and clearance, leading to aggregate formation. Since the effect of Aβ aggregation is neurotoxic and, consequently, related to neuroinflammation and neuronal dysfunction in AD patients, finding effective drugs that are capable of preventing Aβ aggregation is of great interest [97]. A number of marine-derived compounds, belonging to different chemical classes, have been shown to inhibit or decrease Aβ aggregation.

### 4.1. Alkaloids

Lodopyridone A (**116**) (Figure 9), isolated form a marine-derived actinomycete *Saccharomonospora* sp. CNQ-490, was found to reduce a production of Aβ_40_ and Aβ_42_ in SH-SY5Y cells up to 41% and 49% at 1 mM, when compared to a reference standard, isoliquiritigenin, which decreased the Aβ_40_ and Aβ_42_ levels to 48% and 50%, respectively, at 51.5 µM [73].

Liu et al. examined a neuroprotective effect of the bis-indole alkaloids, racemosins A (**173**) and B (**174**), and caulerpin (**175**) (Figure 14), obtained from the EtOAc fraction of the crude aqueous ethanol extract of a green alga *Caulerpa racemose*, which was collected from the Zhanjiang coastline in the East China Sea, by using Aβ_25–35_-induced neurotoxicity in SH-SY5Y cells assay-guided isolation. Compound **173** exhibited neuroprotection with 14.6% increase in cell viability at a concentration of 10 μM, whereas **174** and **175** showed weak to moderate neuroprotective activity with 5.5% and 8.1% increase in cell viability at a concentration of 10 μM, respectively. The positive control, epigallocatechin gallate (EGCG), showed 16.57% increase in cell viability at 10 μM [98].

### 4.2. Phenolics

Xanthocillin X dimethyl ether (**176**) (Figure 15) was obtained from the culture of a marine sponge-associated fungus *Dichotomomyces cejpii,* isolated from a sample of a marine sponge *Callyspongia* sp. cf. *C. flammea*, which was collected at Bare Island, Sydney, Australia. Compound **176** was evaluated for its capacity to decrease a production of Aβ_42_ in N2a cells, stably transfected with human APP695 (N2a-APP695 cells) induced by aftin-5. It was found that cells solely treated with 100 µM aftin-5 displayed a fold change (the amount of Aβ peptides produced by treated cells compared with the Aβ peptides produced by untreated cells) of ±9.4, whereas a pretreatment with 10 µM of **176** reduced the fold of change to ±2.9. A positive control *N*-[*N*-(3,5-difluorophenylacetyl)-L-alanyl-]-(*S*)-phenylglycine*-t*-butylester (DAPT), a known inhibitor of Aβ_42_ production, reduced the fold change to ±0.3 [99].

Phlorotannins, including **61**, **63**, dioxinodehydroeckol (**177**), dieckol (**178**), and phlorofucofuroeckol-A (**179**) (Figure 15), isolated form the EtOAc fraction of the ethanol crude extract of *E. stolonifera*, were evaluated for their capacity to inhibit Aβ_25–35_ self-aggregation by an in vitro ThT assay. Compound **179** exhibited the strongest inhibitory effect, with 80.00% inhibition, followed by **178**, **177**, and **63**, with percent inhibitions of 66.98%, 66.07%, and 34.45%, respectively. However, **61** showed no inhibitory effect on Aβ_25–35_ self-aggregation even at a concentration of 50 µM. Moreover, **63**, **177**, **178**, and **179** showed dose-dependent inhibitory effects on Aβ_25–35_ self-aggregation with IC_50_ values in the range of 6.18 to 34.36 µM, especially **177**, **178**, and **179** exhibited lower IC_50_ values (6.18, 7.93, and 8.31 µM, respectively) than the reference compound, curcumin (IC_50_ = 10.73 µM). Molecular docking and MD simulation analyses confirmed that these phlorotannins have a strong potential to interact with Aβ_25–35_ peptides and interrupt their self-assembly and conformational transformation, thereby inhibiting Aβ_25–35_ aggregation. Interestingly, the docking study revealed that bulky compounds of more than three repeating phloroglucinol units interacted evenly with most of Aβ_25–35_ residues. However, the phloroglucinol monomer is not expected to cause structural changes to Aβ_25–35_ because of its very small structure, although it did form hydrogen bonds with Gly-25, Asn-27, and Ile-32 residues of the peptide [100].

### 4.3. Terpenoids

#### 4.3.1. Meroterpenoids

An α-tocopherol derivative, α-tocospirone (**180**) (Figure 16), isolated from the EtOAc-soluble fraction of the crude aqueous ethanol extract of a green alga *Caulerpa racemosa*, collected from the Zhanjiang coastline in the East China Sea, displayed a significant effect against Aβ_25–35_-induced cell damage in SH-SY5Y cells with 13.6% increase in cell viability at 10 μM [101].

#### 4.3.2. Steroids and Their Derivatives

16-*O*-Desmethylasporyergosteron-β-d-mannoside (**181**) (Figure 16), isolated from the EtOAc extract of a marine sponge-associated fungus *Dichotomomyces cejpii*, was found to reduce the Aβ_1–42_ production by N2a-APP695 cells induced by aftin-5 with the fold change to ±3.8, at 10 µM, whereas the positive control, DAPT, reduced the fold change to ±0.3, respectively [99]. (23*E*)-3β-Hydroxy-stigma-5,23-dien-28-one (**182**) and (22*E*)-3β-hydroxy-cholesta-5,22-dien-24-one (**183**) (Figure 16), isolated from the EtOAc-soluble fraction of the crude extract of a green alga *Caulerpa racemosa*, showed significant neuroprotective effects towards Aβ_25–35_-induced SH-SY5Y cell damage with 11.3% and 16.0% increase in cell viability at a concentration of 10 μM, respectively [101].

#### 4.3.3. Tetraterpenes

Compound **88** (Figure 7), extracted from a seaweed *S. horneri,* was found to inhibit Aβ_1–42_ fibrils formation by ThT assay. Interestingly, **88** exhibited a greater potency of inhibition of the formation of Aβ_1–42_ fibrils when compared with curcumin, an inhibitor of Aβ fibrils formation. Compound **88,** at 0.1–1 μM, substantially reduced the formation of Aβ_1–42_ oligomers in a dot blotting assay. Moreover, small amounts of irregular aggregates were observed in the sample after coincubation of **88** and Aβ_1–42_, suggesting that **88** largely altered the shape of Aβ_1–42_ assemblies. Moreover, the cell viability of SH-SY5Y cells decreased to ca. 50% by treatment with 1 μM of **88**-modified Aβ_1−42_ oligomers, suggesting that **88**-modified Aβ_1−42_ oligomers were less toxic than Aβ_1−42_ oligomers to SH-SY5Y cells. Analysis of direct interactions between **88** and Aβ_1−42_ peptide, based on the trajectories of all-atom MD simulations, reveals that nine molecules of **88** clustered with each other, bound to Aβ_1−42_ peptide, and formed stable aggregates, which could inhibit the conformational transition of Aβ_1−42_ and a subsequent aggregation. Moreover, analysis of the atomic contacts between molecules of **88** and Aβ_1−42_ monomers indicated that **88** directly bound to Aβ_1−42_ peptide mainly via hydrophobic interactions [102].

### 4.4. Fatty Acids

Since monocyte-derived perivascular macrophages efficiently phagocytose accumulations of Aβ which is involved in a clearance of Aβ plaques, Yuan et al. investigated the effects of DHA (**1**) (Figure 1) on monocytes. They have found that Aβ_25–35_ has a “hormesis” effect on human leukemia monocytic (THP-1) cells viability and necrosis, i.e., a biphasic dose-response relationship (a low dose stimulation and a high dose inhibition). The results showed that pretreatment with **1** significantly inhibited the activation of THP-1 cells induced by Aβ_25–35_ in a concentration-dependent manner as well as inhibition of the anti-migratory effect caused by Aβ_25–35_ on THP-1 monocytes. Compound **1** also suppresses Aβ-induced-expression of pro-inflammatory cytokines expression, namely TNF-α, IL-1β, and IL-6. Compound **1** can also prevent Aβ-induced necroptosis of THP-1 cells via the receptor interacting protein kinase-1 (RIPK1)/RIPK3 signaling pathway, indicating that treatment with **1** restored migration of THP-1 monocytes induced by Aβ_25–35_. This suggests that treatment with **1** could be promising for AD management [103].

### 4.5. Peptides

Li et al. isolated nine oligopeptides from a sea cucumber (*Stichopus japonicas*) hydrolysates, prepared by simulated gastrointestinal digestion (pepsin-pancreatin system) by gel permeation (Sephadex G-50) and ion exchange (DEAE-52 cellulose resins) column chromatography. The nine peptides were purified by reversed-phase ultra-high-performance chromatography (RP-UPLC) coupled to an ion trap MS, and whose sequences were identified as GMR (Gly-Met-Arg), DVE (Asp-Val-Glu), VFP (Val-Phe-Pro), LGFH (Leu-Gly-Phe-His) LGFH (Ile-Gly-Phe-His), FQF (Phe-Gln-Phe), LCK (Leu-Cys-Lys), ICK (Ile-Cys-Lys), and DWF (Asp-Trp-Phe). Molecular docking study indicated that IGFH, LGFH, DWF, and FQF showed promising CD38 inhibitory activity. These four peptides (0.1 and 0.5 mM) also exhibited significant anti-Aβ aggregation activity in a mCherry-Aβ (E22G) cells model, when compared to the model group [104].

### 4.6. Carbohydrates

Acidic oligosaccharide sugar chain (AOSC) is a marine-derived acidic oligosaccharide rich in mannuronate building blocks with the average molecular weight of 1300 Da. Pretreatment of SH-SY5Y cells in the presence of aged Aβ_25–35_ or 2 µM of aged Aβ_1–40_ with 50 and 100 µg/mL of AOSC, isolated from a brown alga *Ecklonia kurome* OKAM by enzymatic depolymerization for 24 h, was able to attenuate Aβ-induced inhibition of MTT reduction, and the maximal inhibitory effect of AOSC was observed at 100 µg/mL. AOSC reduced the inhibition of MTT reduction of primarily cultured cortical cells induced by aged Aβ_25–35_. Additionally, pretreatment with 50 and 100 µg/mL of AOSC for 24 h was found to suppress the Aβ_25–35_-induced apoptosis [105].

### 4.7. Miscellaneous

Homotaurine (3-amino-1-propanesulfonic acid) (**6**) (Figure 1) is a naturally occurring metabolite of various marine red algae [106,107]. Compound **6** has been considered as a potential treatment for AD as it binds to soluble Aβ and decreases Aβ_42_-nduced cell death in neuronal cell culture [108].

## 5. Marine-Derived Compounds That Inhibit Protein Kinases

Inhibition of either protein kinase C*θ* (PKC*θ*) or protein kinase C*δ* (PKC*δ*) could effectively reduce Aβ levels and reverse AD phenotypes [109]. In addition, targeting the serine/threonine protein kinase, glycogen synthase kinase 3β (GSK-3β), is an important clue in several signaling pathways, such as cell proliferation, apoptosis, inflammation, and glycogen metabolism, which are involved in treatment of AD, Parkinson’s disease and type 2 diabetes mellitus [110]. The most significant approach that has been recently pursued is multitarget-directed ligands (MTDLs), including the GSK-3β enzyme. Indeed, protein kinases and phosphatases control tau phosphorylation, and any imbalance between them caused the tau phosphorylation and consequently aggregation [111]. Several classes of marine-derived compounds are found to be PKC inhibitors.

### 5.1. Alkaloids

Biscogniauxone (**184**) (Figure 17), an isopyrrolonaphthoquinone obtained from the methanol extract of the culture of a marine-derived fungus *Biscogniauxia mediterranea* strain LF657, isolated from a sediment collected in the Eastern Mediterranean Sea (2800 m depth), was found to inhibit GSK-3β, with an IC_50_ value of 8.04 µM. The positive control, 4-benzyl-2-methyl-1,2,4-thiadiazolidine-3,5-dione (TDZD-8) showed an IC_50_ value of 0.26 µM [69].

Plisson et al., in their screening program of a collection of approximately 2600 Southern Australian and Antarctic marine invertebrates and algae, for the ability to inhibit CK1d, CDK5, and GSK-3β, combined with chemical (HPLC–DAD–MS) and spectroscopic (^1^H NMR) profiling of the extracts that inhibited these protein kinases, have identified an ascidian *Didemnum* sp. (CMB-02127), collected off the Northern Rottnest Shelf, Western Australia, as a high potential extract. This extract furnished several members of pyrrole-polyphenol alkaloids, including ningalins A–G (**185**–**191**) and lamellarins Z (**192**), G (**193**), and A6 (**194**) (Figure 17). Compounds **186**–**191**, and to a lesser extent, **192**–**194**, exhibited a range of kinase inhibitory activities. Compounds **186**, **189,** and **190** were modest inhibitors of casein kinase 1δ (CK1δ) and GSK-3β (IC_50_ = 0.8–3.9 μM). Curiously, only **186** showed inhibition of CDK_5_ (IC_50_ = 2.6 μM). On the contrary, **192**–**194** did not inhibit CK1δ but were modest inhibitors of CDK5 (IC_50_ = 1.0–5.6 μM), with only **192** that inhibited GSK-3β (IC_50_ = 3.0 μM). Molecular docking of **186**–**194** into the catalytic site of the previously reported co-crystal of kinase CDK5^D144n^/p25 with aloisine predicted that the ningalins preferentially bind in the ATP binding site. All ligands are predicted to share common hydrogen bond interactions to the hinge region, i.e., Cys-83, Asp-84, and Gln-85, and to the Asn-144 terminal amide. The molecular model also predicted that the orthogonal aromatic rings of **186**, **187,** and **189** are involved in additional hydrophobic interactions. Compound **190** and **191** are predicted to specifically form hydrogen bonds with Glu-81 and Tyr-15. Compounds **192**–**194** are predicted for their preference to bind in the ATP binding pocket. However, the hydroxyl and steric methoxy groups were pointing in the opposite direction to the hinge region (Cys-83) and few hydrogen bonds were formed with the protein backbone [112].

Hymenialdisine (**195**) (Figure 18) was first isolated in 1982 from the marine sponges *Axinella verrucosa* and *Acanthella aurantiaca* [113]. Further investigation showed that **195** is a cyclin-dependent kinases inhibitor, especially toward GSK-3β and CK1 (IC_50_ = 10 and 35 nM, respectively). Compound **195** also blocked the phosphorylation of tau protein at sites that are hyperphosphorylated by GSK-3β and CDK5/p35 in AD. Moreover, the inhibitory activity of **195** was examined against CDK1/cyclin B, CDK2/cyclin A, CDK2/cyclin E, CDK3/cyclin E, and CDK5/p35, in the presence of 15 μM ATP, presenting IC_50_ values of 22, 70, 40, 100, and 28 nM, respectively. Meijer et al. have also found that the other hymenialdisine derivatives, including axinohydantoin (**196**), dibromocantharelline (**197**), and hymenidin (**198**) (Figure 18) are able to inhibit the special kinases, i.e., CDK1/cyclin B, CDK5/p35, GSK-3β, and CK1, which were moderately sensitive to **196**, with IC_50_ values ranging from 3 to 7 μM. On the other hand, **197** and **198** selectively inhibited GSK-3β and CDK5/p35, with IC_50_ values of 3 and 4 μM, respectively [114]. Plisson et al. have isolated several bromopyrrole alkaloids, including from the extract of a marine sponge *Callyspongia* sp. (CMB-01152) which displayed inhibitory activity against the neurodegenerative disease kinase targets: CK1, CDK5, and GSK-3β. However, the kinase inhibitory activity detected in the sponge extract was found to be due to **195**, a major metabolite whose IC_50_ values against kinases were 0.03 μM for CK1δ, 0.16 μM for cyclin-dependent kinase (CDK5/p25), and 0.07 μM for GSK-3β [115]. Leucettamine B (**199**) (Figure 18) is a marine-derived imidazole alkaloid, first isolated from a sponge *Leucetta microraphis* Haechel, collected from Argulpelu Reef in Palau at a depth of 30 m [116]. Debdabe et al. synthesized **199** and its derivatives (leucettines) and assayed them on eight purified kinases, including tyrosine phosphorylation regulated kinases (DYRK1A and DYRK2), cdc2-like kinases (CLK1 and CLK3), CDK5/p25, CK1, GSK-3β, and Pim1. Compound **199** inhibited DYRK1A, DYRK2, CLK1, and CLK3, with IC_50_ values of 2.8, 1.5, 0.40, and 6.4 μM, respectively, but did not show any inhibitory activity (IC_50_ > 10 μM) against CDK5/p25, CK1, GSK-3β and Pim1. Among **199** derivatives, leucettine L_41_ (**200**) (Figure 18) exhibited the highest inhibitory activities toward DYRK1A (IC_50_ = 0.040 μM), DYRK2 (IC_50_ = 0.0.35 μM), CLK1 (IC_50_ = 0.0.15 μM), CLK3 (IC_50_ = 4.5 μM), GSK-3β (IC_50_ = 0.41 μM), and Pim1 (IC_50_ = 4.1 μM) [117]. Furthermore, Tahtouh et al., in their study to compare the activity of **199** and **200** with harmine (a reference DYRK1A inhibitor) against a panel of 26 kinases to confirm the potential therapeutics of leucettines against neurodegenerative diseases such as AD, have found that **200** was a potential kinases inhibitor. Interestingly, **200** not only showed a neuroprotective potency on glutamate-induced HT22 cell death, but also reduced the APP-induced cell death in cultured rat brain slices [118].

Manzamine A (**201**), a β-carboline alkaloid first isolated from a marine sponge *Haliclona* sp. in 1986, is responsible for a myriad of bioactivities [119]. Hamman et al. isolated **201**, along with its derivatives, 6-hydroxymanzamine A (**202**), 8-deoxymanzamine A (**203**), manzamine E (**204**), and manzamine F (**205**) (Figure 18), from an Indonesian marine sponge *Acanthostrongylophora* sp. Inhibition studies of **201** against a selected panel of five different kinases related to GSK-3β, specifically CDK1, protein kinase A (PKA), CDK-5, mitogen-activated protein kinase (MAPK), and GSK-3α, revealed that **201** inhibited specifically GSK-3β (IC_50_ = 10.2 μM) and CDK-5 (IC_50_ = 1.5 μM), the two kinases involved in tau hyperphosphorylation. Kinetic study suggested that **201** acts as a noncompetitive inhibitor of ATP binding because increasing ATP concentration does not interfere with the inhibition. Interestingly, when carboline and ircinal A, the precursors of **201**, were tested for their potential GSK-3β inhibition, both moieties were inactive, indicating that the entire manzamine molecule is required for this activity. Moreover, treatment of the SH-SY5Y cell culture with **201,** at different concentrations (5, 15, and 50 µM), resulted in a decrease in tau phosphorylation [120].

### 5.2. Phenolics

Aromatic butenolides, namely eutypoids B–E (**206**–**209**) (Figure 19) were isolated from the EtOAc extract of the culture of a marine-derived fungus *Penicillium* sp. KF620, isolated from the water sample of the North Sea. Compounds **206**–**209** showed moderate inhibitory activity against GSK-3β with IC_50_ values of 0.78, 0.67, 4.08, and 2.26 µM, respectively, compared to the positive control, SB-415286 (**210**), whose IC_50_ value is 0.09 µM [121]. Wiese et al. described the isolation of three highly oxygenated benzocoumarins, pannorin (**211**) from a fermentation broth extract of *Aspergillus* sp. strains LF666, isolated from a deep-sea sediment sample collected from the Mediterranean Sea, alternariol (**212**), and alternariol-9-methyl ether (**213**) (Figure 19) from the mycelium extract of *Botryotinia fuckeliana* KF666, obtained from the German Wadden Sea. In vitro luminescent assay for GSK-3β inhibitory activity of **211**–**213** showed that **212** exhibited the strongest inhibitory activity, with IC_50_ = 0.13 μM, while **213** and **213** showed weaker activity, with IC_50_ values of 20 and 0.35 μM, respectively (the positive control TDZD-8, a non-ATP competitive inhibitor, exhibited IC_50_ = 0.26 μM) [122].

A phlorotannin, eckmaxol (**214**) (Figure 19), obtained from the EtOAc fraction of a crude aqueous ethanol extract of a South African brown macroalga *Ecklonia maxima* showed an ability to produce anti-Aβ plaque neuroprotective effects in SH-SY5Y cells by preventing Aβ peptide-induced neuronal apoptosis, and an increase in ROS. Moreover, **214** significantly activated other mechanisms, including an increase in expression of phosphor-extracellular signal-regulated kinase along with a decrease in expression of phosphor-Ser9-GSK-3β. Molecular docking simulation of **214** indicated a good affinity in the ATP binding site of GSK-3β and MAPK, suggesting that **214** could be an excellent candidate for anti-AD agent [123]. A preliminary screening for neuroprotective properties of a Southern Australian brown alga *Zonaria spiralis* revealed a promising inhibitory activity towards BACE-1, CDK5, CK1δ, and GSK-3β. Bioactivity-guided purification led to the isolation of phloroglucinol-derived hemiketal lipids, namely spiralisones A (**215**), B (**216**), C and D, a degraded fucoxanthin metabolite (apo-9ʹ-fucoxanthinone), together with 5,7-dihydroxy-2-tridecanyl chromone and 5,7-dihydroxy-2-(4*Z*-7*Z*-10*Z*-13*Z*-16*Z*-nonadecapentaenyl)-chromone (**217**) (Figure 19) from a dichloromethane-soluble fraction. Compounds **215**–**217** inhibited not only CDK5/p25, with IC_50_ values of 10, 3 and 10 μM, respectively, but also CK1δ and GSK-3β, with IC_50_ values < 10 μM [124].

### 5.3. Terpenoids

#### Sesquiterpenes

Palinurin (**218**) (Figure 20) is a linear furanosesquiterpene, isolated from a Mediterranean sponge *Ircinia dendroides.* Both naturally occurring and synthetic version of **218** were found to dose-dependently inhibit GSK-3β in human neuroblastoma cells SH-SY5Y, with IC_50_ values of 2.6 mM and 1.9 mM, respectively. Kinetic study revealed its noncompetitive inhibition. Moreover, GSK-3β inhibition by **218** can be competed neither by ATP nor peptide substrate, therefore strongly suggesting that the binding site for **218** lays outside the binding pockets for both ATP and the substrate. A docking study and MD simulations suggested that **218** exerts its inhibitory activity through binding to an allosteric site located at the *N*-terminal lobe of the enzyme. Binding of **218** to this unconventional site modulates the accessibility of ATP γ-phosphate by constraining the conformation of the glycine-rich loop. The authors proposed that this novel allosteric mechanism of action confers **218** a high degree of selectivity towards GSK-3β with respect to other kinases, making it an interesting candidate molecule for the development of new selective and more potent drug for the treatment of GSK-3β-mediated diseases [125].

### 5.4. Naphthoquinones

Adociaquinone B (**219**) and secoadociaquinone B (**220**) (Figure 20), two naphthoquinone derivatives isolated from a crude extract of a marine sponge *Xestospongia* sp., collected in North Sulawesi, Indonesia, were evaluated, together with helenaquinol sulfate (**221**) and xestosaprol C methylacetal (**222**) (Figure 20), isolated from *X. testudinaria,* which was collected in South Pacific, for their inhibitory activity toward several protein kinases. Compounds **219** and **220** exhibited modest but selective inhibitory activities towards CDK9/cyclin T (IC_50_ = 3 µM) and CDK5/p25 (IC_50_ = 6 µM), respectively. Compound **221** showed significant activity against most protein kinases tested (CDK1, CDK2, CDK5, CDK9, CK1, CLK1, DYRK1A, and GSK-3β) with IC_50_ values ranging from 0.5 to 7.5 µM, whereas **222** only showed marginal activity against DYRK1A [126].

### 5.5. Miscellaneous Marine Natural Products

The culture filtrate of a fermentation of a marine member of *Streptomyces griseus* 16S rRNA clade, strain NTK 935, which was isolated from a deep-sea sediment collected from the Canary Basin, furnished 1,4-benzoxazine-type metabolite, named benzoxacystol (**223**) (Figure 20). Compound **223** showed an in vitro inhibitory activity against GSK-3β with IC_50_ value of 1.35 μM [127].

Carteriosulfonic acids A–C (**224**–**226**), metabolites containing 4,6,7,9-tetrahydroxylated decanoic acid subunit that forms an amide linkage with taurine and further esterified at O-9 with long-chain hydroxy unsaturated fatty acids, were isolated from the methanol extract of a marine sponge *Carteriospongia* sp. (PSO-04-3-79), collected in San Miguel Island, Sorsogon, Philippines. Compounds **224**–**226** (Figure 20) inhibited GSK-3β activity with IC_50_ values of 12.5, 6.8, and 6.8 μM, respectively. The impact of long-chain hydroxy unsaturated fatty acids against GSK-3β activity was investigated by desacyl-carteriosulfonic acid (**227**). Interestingly, no significant GSK-3β inhibitory activity was observed at concentrations up to 50 μM, indicating that the long-chain hydroxy unsaturated fatty acid component is necessary for GSK-3β inhibition [128].

## 6. Marine-Derived Compounds with Miscellaneous Enzyme Inhibition

An anabaseine-like derivative, 3-[(2,4-dimethoxy]benzylidene]-anabaseine (GTS-21; **117**), isolated from a nemertean (ribbon worm, *Amphiporus* sp.), has been reported to exhibit an agonist activity on α7 nicotinic acetylcholine receptors. In addition to suppression of the γ-secretase activity in vitro and in vivo, **117** was found to not only promote microglial Aβ phagocytosis in APdE9 mice but also ameliorate cognitive impairment in a mouse model of AD [75]

In recent years, great effort has been made to unravel new enzymes and pathways, which are involved in AD. One group of these enzymes is glutaminyl cyclases (QCs), which belong to the class of acyl transferases. QCs could be associated with pathophysiological processes of various diseases including AD [129]. Evidences revealed that QCs are responsible for the formation of pGlu-modified Aβ peptides in AD, which are more neurotoxic, hydrophobic, and resistant to aminopeptidase degradation compared to unmodified Aβ peptides and, thus, accumulating in AD patients’ brains [130] Therefore, inhibition of QCs may have therapeutic potential for treatment of disorders associated with protein aggregation and (neuro) inflammation and, therefore, might be considered as one of therapeutic approaches in treatment of AD [131]. In this context, Hielscher-Michael et al. evaluated the QCs inhibitory activity of the sulfolipids, 1,2-di-*O*-palmitoyl-3-*O*-(6ʹ-deoxy-6ʹ-sulfo-D-glycopyranosyl)-glycerol (**228**), 1-*O*-palmitoyl-2-*O*-linolenyl-3-*O*-(6ʹ-deoxy-6ʹ-sulfo-D-glycopyranosyl)-glycerol (**229**), and 1-*O*-linolenyl-2-O-palmitoyl-3-*O*-(6ʹ-deoxy-6ʹ-sulfo-D-glycopyranosyl)-glycerol (**230**), isolated form a microalga *Scenedesmus* sp. by using a new reverse metabolomics approach. Compounds **228**–**230** displayed QCs inhibition ranging from 76% to 81% at concentrations of 0.025 and 0.25 mg/mL, respectively [132].

It is important to mention also that during a protein glycation process, a kind of compounds, named advanced glycation end products (AGEs), are generated. Accumulation of AGEs is found in the brain of the aging and AD patients [133]. For this reason, the potency of a seaweed *Fucus vesiculosus* fractions and its purified phlorotannins were investigated for their capacity to prevent protein glycation using two methods viz. Bovine Serum Albumin (BSA)-methylglyoxal and BSA-glucose assays. In both assays, the EtOAc fraction showed a promising activity with EC_50_ value of 0.169 and 0.278 mg/mL, respectively. Further purification of the active fraction led to the isolation of fucophlorethol (**231**), tetrafucol (**232**), and trifucodiphlorethol (**233**) (Figure 21) [134].

## 7. Future Perspectives and Conclusions

This review highlights the importance of marine organisms as a new source of compounds with promising anti-AD activities. The search for new sources of marine-derived compounds with a capacity to cure or prevent neurodegenerative diseases is very crucial, as a number of patients suffering from these diseases is rapidly increasing due to the aging population, especially in highly developed countries. Overall, we have listed 233 marine-derived compounds that possess biological and pharmacological activities, which are considered to be relevant for therapeutic targets for AD. Among the compounds discussed, 102 exhibited anti-AChE and anti-BuChE activities, 68 showed anti-BACE-1 activity, 17 showed anti-Aβ aggregation, 44 inhibited protein kinases activities, and 7 are inhibitors of other enzymes related to AD. Although a significant number of inhibitors of enzymes implicated in AD pathogenesis has been discovered during the past decade, a broader range of therapeutics is still needed. Given the diverse pathology of AD, targeting only the key enzymes could not be the most effective way to treat AD. Interestingly, the most significant approach that has been recently pursued is multitarget-directed ligands (MTDLs), targeting the GSK-3β enzyme with multiple mechanisms. This goal can be achieved by marine natural products due to their structural uniqueness and their unprecedented mechanisms of actions, which are not found in synthetic and terrestrial natural products. In summary, marine-derived compounds can be potential candidates for the development of therapeutic arsenals for AD, per se, or can be explored as unique and unprecedented scaffolds for medicinal chemistry research to discover potent and effective therapeutics for AD.

## Figures and Tables

**Figure 1 marinedrugs-19-00410-f001:**
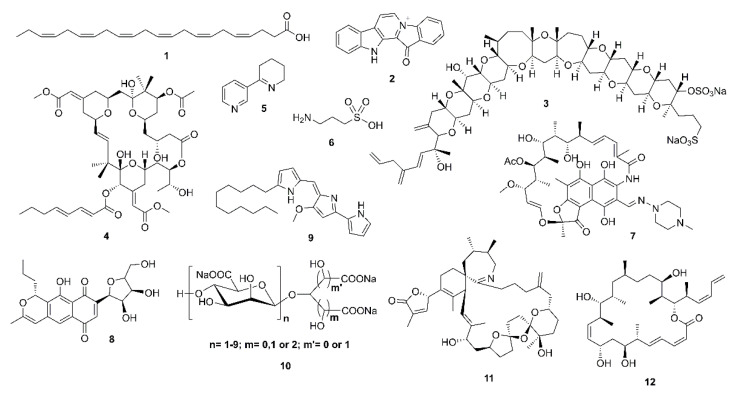
Chemical structures of **1**–**12**.

**Figure 2 marinedrugs-19-00410-f002:**
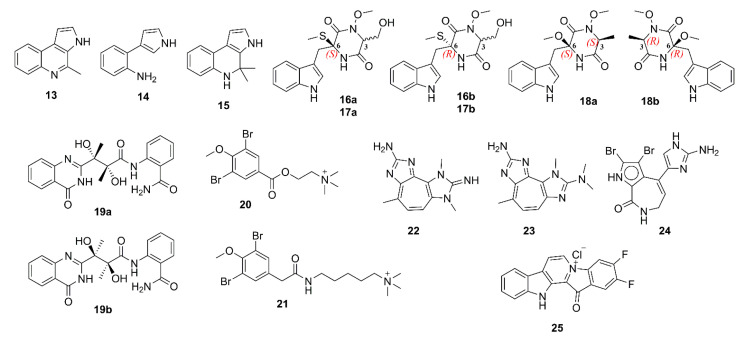
Chemical structures of **13**–**25**.

**Figure 3 marinedrugs-19-00410-f003:**
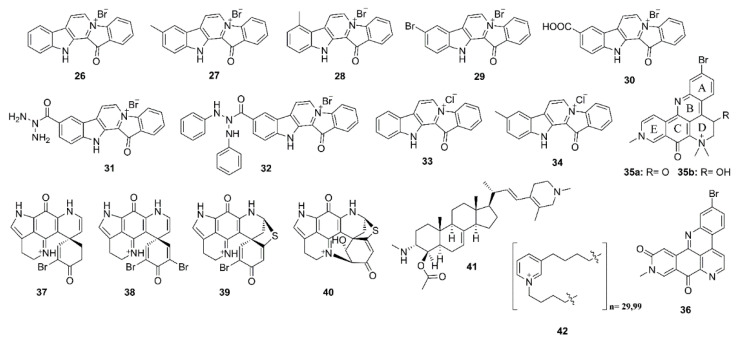
Chemical structures of **26**–**42**.

**Figure 4 marinedrugs-19-00410-f004:**
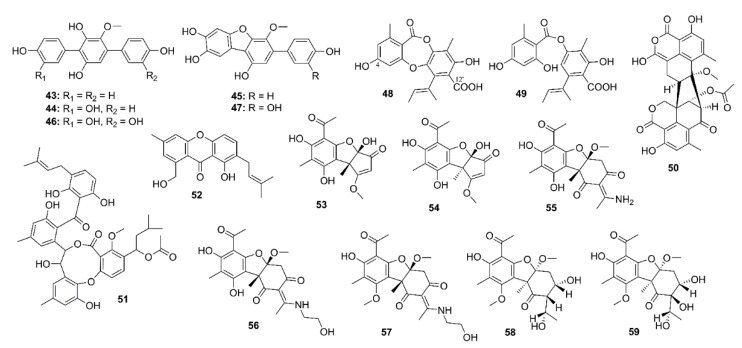
Chemical structures of **43**–**59**.

**Figure 5 marinedrugs-19-00410-f005:**
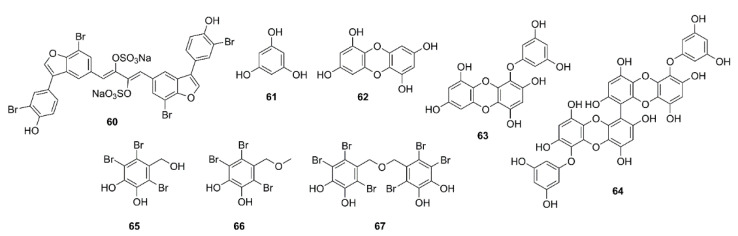
Chemical structures of **60**–**67**.

**Figure 6 marinedrugs-19-00410-f006:**
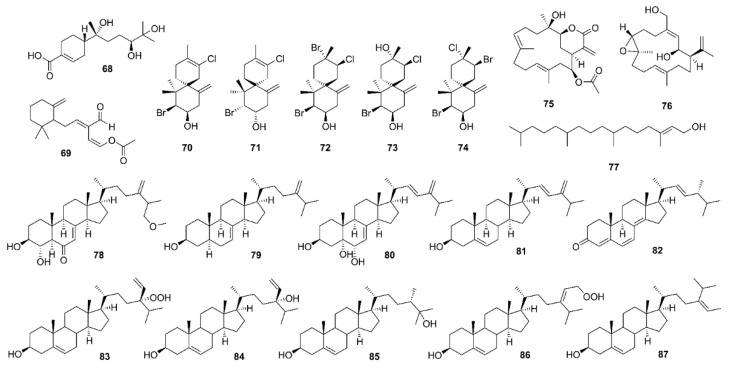
Chemical structures of sesquiterpenes (**68**–**74**), diterpenes (**75**–**77**), and sterols (**78**–**87**).

**Figure 7 marinedrugs-19-00410-f007:**
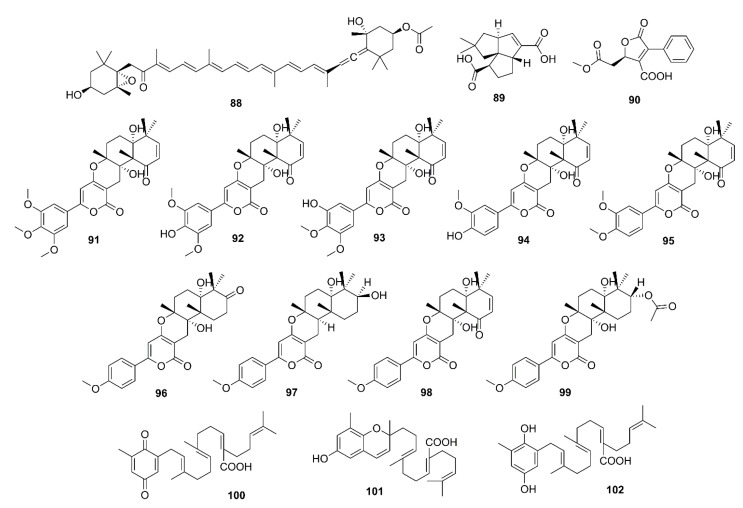
Chemical structures of a tetraterpene (**88**) and meroterpenoids (**89**–**102**).

**Figure 8 marinedrugs-19-00410-f008:**
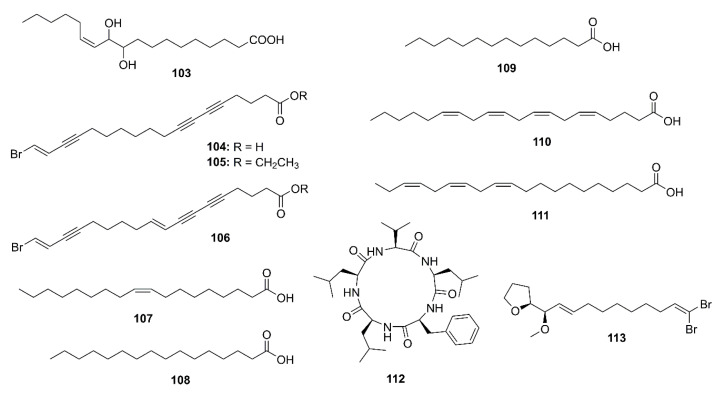
Chemical structures of **103**–**113**.

**Figure 9 marinedrugs-19-00410-f009:**
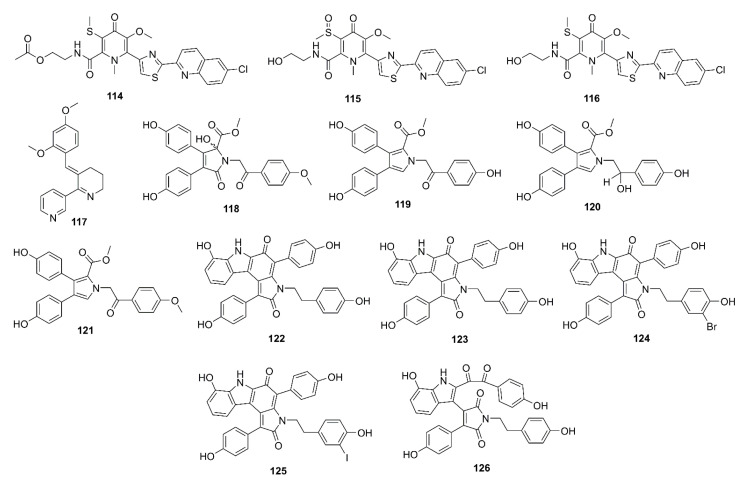
Chemical structures of **114**–**126**.

**Figure 10 marinedrugs-19-00410-f010:**
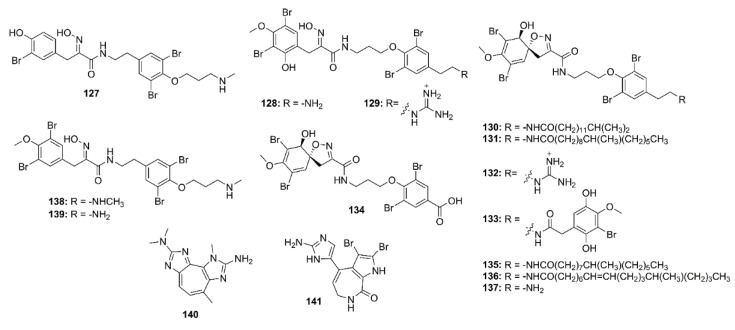
Chemical structures of **127**–**141**.

**Figure 11 marinedrugs-19-00410-f011:**
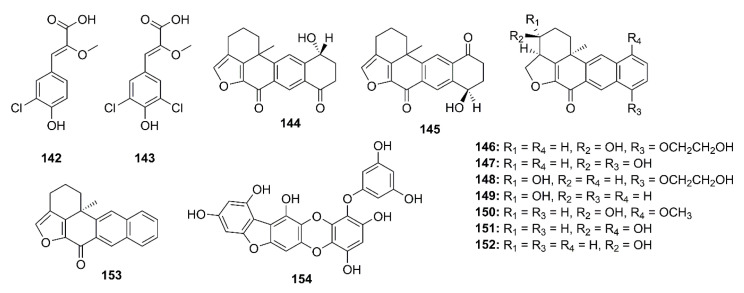
Chemical structures of **142**–**154**.

**Figure 12 marinedrugs-19-00410-f012:**
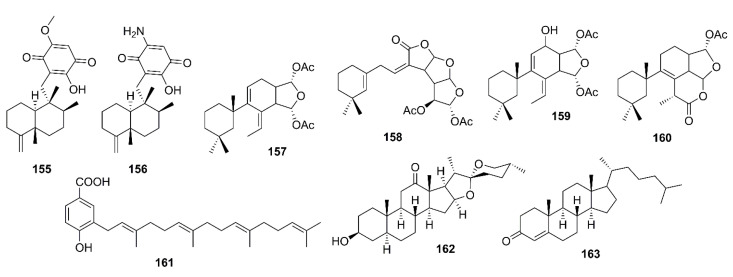
Chemical structures of sesquiterpenes (**155** and **156**), diterpenes (**157**–**160**), meroterpene (**161**), and sterols (**162** and **163**).

**Figure 13 marinedrugs-19-00410-f013:**
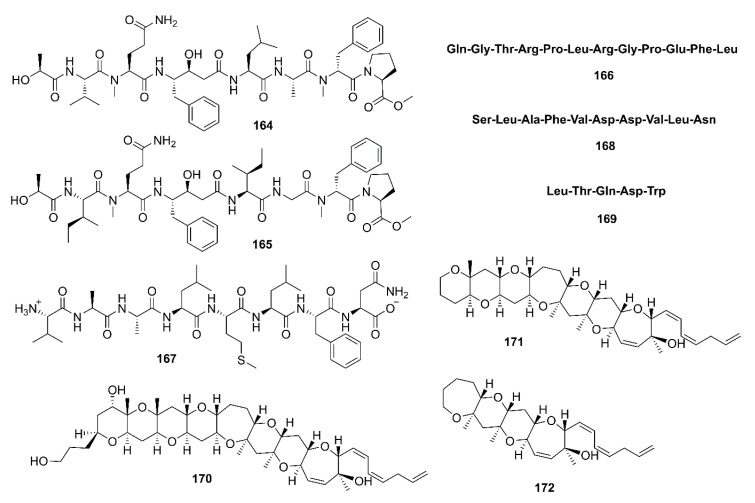
Chemical structures of peptides (**164**–**169**) and polycyclic ethers (**170**–**172**).

**Figure 14 marinedrugs-19-00410-f014:**
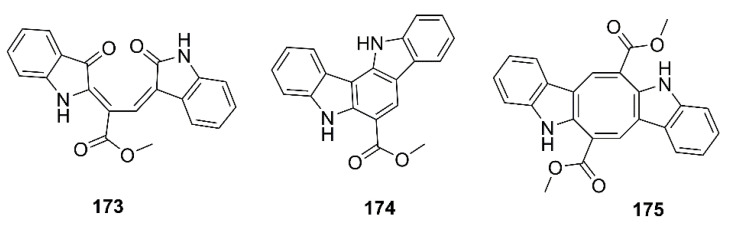
Chemical structures of **173**–**175**.

**Figure 15 marinedrugs-19-00410-f015:**
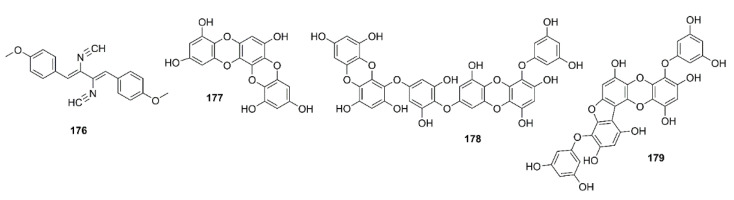
Chemical structures of **176**–**179**.

**Figure 16 marinedrugs-19-00410-f016:**
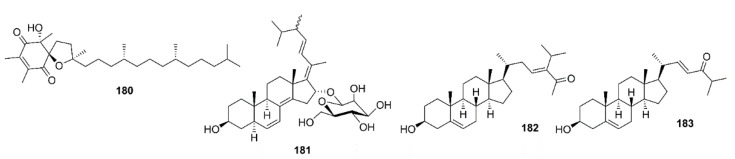
Chemical structures of meroterpene (**180**) and sterols (**181**–**183**).

**Figure 17 marinedrugs-19-00410-f017:**
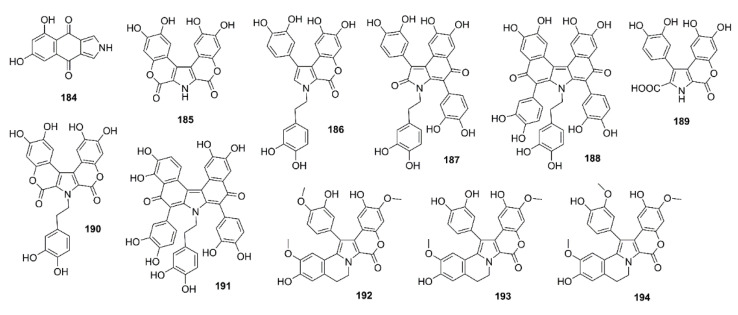
Chemical structures of **184**–**194**.

**Figure 18 marinedrugs-19-00410-f018:**
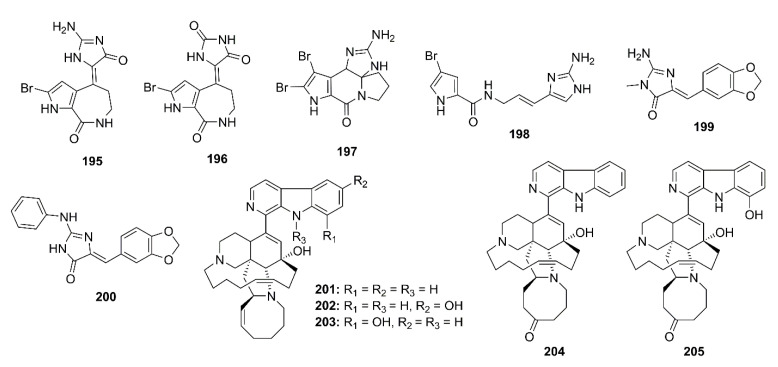
Chemical structures **195**–**205**.

**Figure 19 marinedrugs-19-00410-f019:**
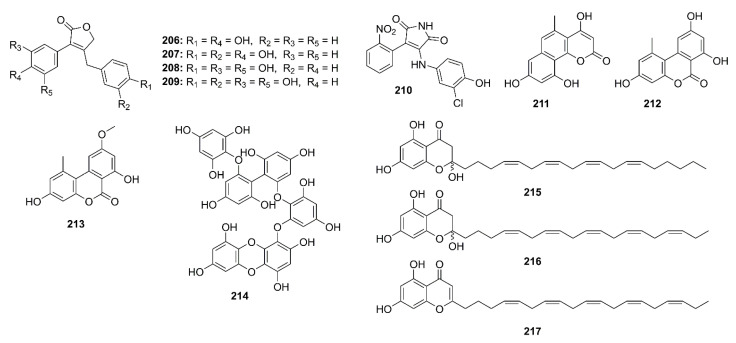
Chemical structures of **206**–**217**.

**Figure 20 marinedrugs-19-00410-f020:**
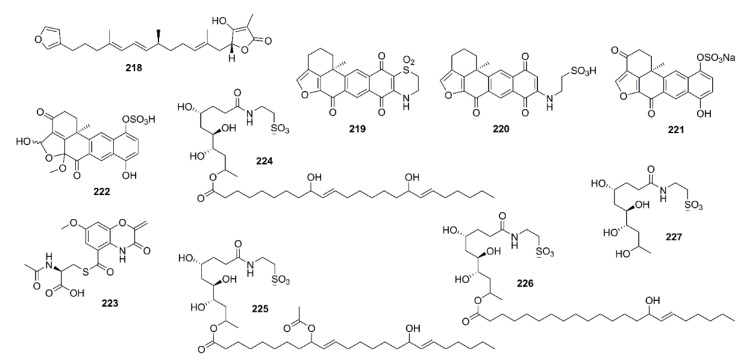
Chemical structures of **218**–**227**.

**Figure 21 marinedrugs-19-00410-f021:**
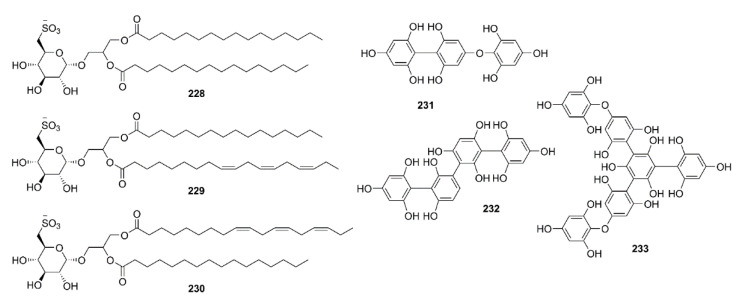
Chemical structures of **228**–**233**.

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
