# Peer review of "Marine-Derived Compounds with Anti-Alzheimer’s Disease Activities"

_marinedrugs, 2021, doi:10.3390/md19080410_

Round 1

Reviewer 1 Report

The manuscript Marine Drugs 1279184, entitled:Marine-Derived Natural Products as Potential Anti-Alzheimer Agents, by S.H. Ghoran and A. Kijjoa is a detailed review of the existing literature on the presence in marine organisms of multiple molecules of potential interest for the treatment of the Alzheimer’s disease.

The value of this review is limited by the large number of similar reviews and the absence of a critical analysis of the data reported and their value both in experimental models and in humans. In particular, because the testing of any molecule in humans is is problematic because of the uncertainty of its therapeutic and toxic effects, some optimistic but unsupported statements in the review should be omitted. Accordingly, the title should avoid to mention “extracts with anti-Alzheimer activities”, as none of the molecules that are mentioned has been so far successful in the treatment of this disease. The review may possibly benefit from an updated introduction to the actual therapy of the Alzheimer’s disease and the future therapeutic perspectives.

OTHER POINTS

 --The name of the pathology is Alzheimer’s Disease, not Alzheimer, since this is the name of the scientist who defined the disease;

--The comparison of the activity of marine extracts to individual molecules should always be based on the same units: g/ml;

--A reference is missing on line 173-174: the arguing is clearly pointing to reference DOI: 10.1080/13880209.2017.1302967,  from the same authors asRef. 30;

--The citation of reviews should not prevail on that of original articles;

--The English language deserves attention.

Author Response

Reviewer # 1:

The manuscript Marine Drugs 1279184, entitled:Marine-Derived Natural Products as Potential Anti-Alzheimer Agents, by S.H. Ghoran and A. Kijjoa is a detailed review of the existing literature on the presence in marine organisms of multiple molecules of potential interest for the treatment of the Alzheimer’s disease.

The value of this review is limited by the large number of similar reviews and the absence of a critical analysis of the data reported and their value both in experimental models and in humans. In particular, because the testing of any molecule in humans is is problematic because of the uncertainty of its therapeutic and toxic effects, some optimistic but unsupported statements in the review should be omitted.

Reply:

We do not agree with reviewer #1’s opinion that “The value of this review is limited by the large number of similar reviews and the absence of a critical analysis of the data reported and their value both in experimental models and in humans. In particular, because the testing of any molecule in humans is problematic because of the uncertainty of its therapeutic and toxic effects, some optimistic but unsupported statements in the review should be omitted”.

First, it is true that several review papers about the potential of marine natural products for AD have recently appeared in several open access journals. This reflects the increasing number of researchers working in this field as well as the importance of marine-derived compounds in the field of neurodegenerative diseases, especially AD. So, if we look at the paper published in 2011 (doi: https://doi.org/10.1039/c0np00027b), we can see that only three marine-derived compounds, i. e. homotaurine, bryostatin-1, and NP-12, have passed to clinical studies. As the interest of “Drugs from the Sea” has been a hot topic since the year 2000 (That is why Marine Drugs is a forefront journal today), the increasing number of marine-derived compounds isolated from marine resources combined with the advancement of in vitro and in vivo assay techniques, has led to the explosion of research in this important area. I am sure that reviewer#1 knows very well that only a small number of natural compounds that showed bioactivity in vitro or even in vivo using animal models reached clinical trial phase in human. However, enzymatic or cell-based assays are very important step for the development of compounds using medicinal chemistry to reach clinical trials. Therefore, the objective of this review is not making a comparison of the bioassay methods or of the animal models with human subject. Instead, as we have stated in our manuscript, the objective of this review is to demonstrate that marine environment, whose biodiversity is well recognized, is a source of structurally unique compounds with a myriad of biological activity and multiple mechanisms can serve as valuable scaffolds for medicinal chemistry for developments of valuable therapeutic drugs in neurodegenerative diseases, especially AD. Besides reporting various classes of compounds with different bioactivities related to ID, we have also discussed the molecular docking as a theoretical support for the experimental results. We have also demonstrated that chemical modifications of many marine-derived compounds have proved that their scaffolds can contribute to the development of therapeutic drugs. Last but not least, contrary to other recent review papers whose scopes are more specific, this review covers a broad range of compounds (223) and several enzymes and targets known to be related to AD.

Reviewer # 1:

Accordingly, the title should avoid to mention “extracts with anti-Alzheimer activities”, as none of the molecules that are mentioned has been so far successful in the treatment of this disease.

Reply

We do agree with the suggestion of reviewer #1 in this respect. As such, we have removed the part of “section 2: Marine Extracts with Anti-Alzheimer Activities” in the first version of the manuscript.

Reviewer # 1:

The review may possibly benefit from an updated introduction to the actual therapy of the Alzheimer’s disease and the future therapeutic perspectives.

Reply

We have updated the introduction part and add some statement in the Future Perspectives and Conclusions.

Reviewer # 1:

OTHER POINTS

 --The name of the pathology is Alzheimer’s Disease, not Alzheimer, since this is the name of the scientist who defined the disease;

Reply:

We have corrected this term.

--The comparison of the activity of marine extracts to individual molecules should always be based on the same units: g/ml;

Reply

We have removed this section as we think the objective of this review should be based on molecules whose molecular structures can be considered as important scaffolds.

--A reference is missing on line 173-174: the arguing is clearly pointing to reference DOI: 10.1080/13880209.2017.1302967, from the same authors asRef. 30;

Reply

This reference belonged to the section of marine extracts. Therefore, it is not included in this revised version.

--The citation of reviews should not prevail on that of original articles;

Reply

As you can see, from all the cited references (133), less than 10 references are review papers.

-The English language deserves attention.

Reply

We have rechecked the English language and have modified the syntax of some sentences. We have also corrected several typos.

Reviewer 2 Report

This manuscript reviews a wide variety of studies on marine extracts as well as purified and characterized marine natural compounds that may serve as leads for development of new Alzheimers therapeutics. Its coverage of details is exhausting, such that it is difficult for the reader to evaluate the relative potential of the different entities described. It would be useful  to include a table listing the most active compounds, as most compounds were only active at very high concentrations. Also, gamma-secretase inhibitors were not included, such as GTS-21, a derivative of the marine toxin anabaseine (See Takata K, et al. 2018 Alpha7 nicotinic acetylcholine receptor-specific agonist DMXBA (GTS-21) attenuates Aβ accumulation through suppression of neuronal γ-secretase activity and promotion of microglial amyloid-β phagocytosis and ameliorates cognitive impairment in a mouse model of Alzheimer's disease. Neurobiol Aging. 62:197-209. 

This reviewer questions the value of detailed reporting of the actions of crude extracts and suggest that these studies be omitted from the review.

Author Response

Reviewer #2

This manuscript reviews a wide variety of studies on marine extracts as well as purified and characterized marine natural compounds that may serve as leads for development of new Alzheimers therapeutics. Its coverage of details is exhausting, such that it is difficult for the reader to evaluate the relative potential of the different entities described. It would be useful  to include a table listing the most active compounds, as most compounds were only active at very high concentrations.

Reply

The objective of this review is to demonstrate that marine environment whose biodiversity is well recognized is a source of structurally unique compounds with a myriad of biological activity and multiple mechanisms as well as the importance of marine-derived compounds in the field of neurodegenerative diseases, especially AD. This is reflected by an increasing number of publications in this field in recent years. Therefore, we do not consider this review is exhaustive. On the contrary, this review is covering as many structural classes of compounds that affect a broader range of enzymes and targets involved in AD. Since the review is first organized by the types of enzymes/targets and then each chemical class of compounds is discussed, this will allow the readers to focus on either targets or classes of compounds. The classification by chemical classes of compounds inside the enzymes/targets in question will allow researchers to pinpoint valuable scaffolds for medicinal chemistry study for the developments of valuable therapeutic drugs in neurodegenerative diseases, especially AD.

Reviewer #2

It would be useful to include a table listing the most active compounds, as most compounds were only active at very high concentrations.

Reply:

We don’t agree with this suggestion because it contradicts reviewer#2’s prior comment. First, this review contains 21 figures of 223 chemical structures. By adding another table, which we think redundant, will even confuse more the readers.

Moreover, as the compounds have been tested in different assays, the IC50 values could not be comparable. Moreover, in each series of tested compounds, we have indicated the reference standards used in the test. As reviewer #2 can see, even the same activity, different authors used different positive controls. Moreover, even the same method, different authors obtained different IC50 values of the same positive controls. Therefore, by adding another table suggested by reviewer # 2 not only makes this manuscript become redundant but also can give a misperception of the reality.

Reviewer #2

Also, gamma-secretase inhibitors were not included, such as GTS-21, a derivative of the marine toxin anabaseine (See Takata K, et al. 2018 Alpha7 nicotinic acetylcholine receptor-specific agonist DMXBA (GTS-21) attenuates Aβ accumulation through suppression of neuronal γ-secretase activity and promotion of microglial amyloid-β phagocytosis and ameliorates cognitive impairment in a mouse model of Alzheimer's disease. Neurobiol Aging. 62:197-209. 

Reply:

We wish to thank reviewer #2’s suggestion. We have add a paragraph about GTS-21 in the body text of the manuscript.

Reviewer #2

This reviewer questions the value of detailed reporting of the actions of crude extracts and suggest that these studies be omitted from the review.

Reply:

We agree with reviewer #2’s suggestion. As such we have removed this section in the revised manuscript.

Round 2

Reviewer 1 Report

The authors have improved the English writing but the overall originality of the review has not been improved. Some important sentences are taken as they appear in the cited reference.